

# Diurnal regulation of photosynthetic light absorption, electron transport and carbon fixation in two contrasting oceanic environments

Nina Schuback[1,2], Philippe D. Tortell[2,3]

[1] Swiss Polar Institute, École Polytechnique Fédérale de Lausanne, Switzerland
[2] Dept. of Earth, Ocean, and Atmospheric Sciences, University of British Columbia, Vancouver, Canada
[3] Dept. of Botany, University of British Columbia, Vancouver, Canada

*Correspondence to*: Nina Schuback (schuback.nina@gmail.com)

**Abstract.** Understanding the dynamics of marine phytoplankton productivity requires mechanistic insight into the non-linear
coupling of light absorption, photosynthetic electron transport and carbon fixation in response to environmental variability.
In the present study, we examined the variability of phytoplankton light absorption characteristics, light-dependent electron
transport and [14]C-uptake rates over a 48 hour period in the coastal Subarctic NE Pacific. We observed an intricately
coordinated response of the different components of the photosynthetic process to diurnal irradiance cycles, which acted to
maximise carbon fixation while simultaneously preventing damage by excess absorbed light energy. In particular, we found
diurnal adjustments in pigment ratios, excitation energy transfer to reaction center II (RCII), the capacity for non-
photochemical quenching (NPQ), and the light efficiency ($\alpha$) and maximum rates ($P_{max}$) of RCII electron transport ($ETR_{RCII}$)
and [14]C-uptake. Comparison of these results from coastal waters to previous observations in offshore waters of the Subarctic
NE Pacific provided insight into the effects of iron limitation on the optimization of photosynthesis. Under iron-limiting
conditions, there was a significant reduction of iron-rich photosynthetic units per chlorophyll *a*, which was partly offset by
higher light absorption and electron transport per photosystem II. Iron deficiency limited the capacity of phytoplankton to
utilize peak mid-day irradiance for carbon fixation, and caused an upregulation of photo-protective mechanisms, including
NPQ, and the decoupling of light absorption, electron transport and carbon fixation. Such decoupling resulted in an increased
electron requirement ($\Phi_{e,C}$) and decreased quantum efficiency ($\Phi_C$) of carbon fixation at the iron-limited station. In both
coastal and off-shore waters, $\Phi_{e,C}$ and $\Phi_C$ correlated strongly to NPQ. We discuss the implications of our results for the
interpretation of bio-optical data, and the parameterization of numerical productivity models, both of which are vital tools in
monitoring marine photosynthesis over large temporal and spatial scales.

## 1 Introduction

It is well known that photosynthetic performance and light harvesting characteristics of phytoplankton vary widely across
environmental conditions and seasonal cycles (e.g. Falkowski and Raven, 2007; Geider et al., 2001; Harris, 1986; Kirk,



1994). On physiological scales, these changes can be observed as rapid metabolic adjustments occurring over seconds to hours, while on ecological scales (days to months) they are manifested as phytoplankton species succession. These physiological and ecological responses are ultimately driven by the integrated growth environment experienced by phytoplankton, and the need to optimise the conversion of light energy to carbon biomass, while preventing damage from

super-saturating light. The present study was designed to improve mechanistic understanding of the entire photosynthetic process in marine phytoplankton and its capacity to respond to environmental variability. Such information is necessary to understand and predict ongoing climate impacts associated with changes in nutrient supply, temperature and irradiance levels on marine photosynthetic carbon fixation (e.g. Behrenfeld et al., 2006, 2016; Hoegh-Guldberg and Bruno, 2010; Taucher and Oschlies, 2011).

The photosynthetic process comprises a chain of diverse reactions, leading from light absorption via electron transport to photosynthate (ATP and NADPH) production and carbon fixation (Fig. 1). These reactions, operating on vastly different time scales, are ultimately powered by solar energy and critically dependent on nutrient availability. Variability in surface ocean nutrient concentrations results from physical mixing and biological consumption acting on scales of days to months. By comparison, variability in light intensity occurs over a broader range of time-scales, with rapid transients induced by

atmospheric variability (e.g. cloud cover) and fine-scale mixing, super-imposed on diel and seasonal cycles. Importantly, while light energy is an absolute requirement for the photosynthetic process, excess irradiance, even on short timescales, can lead to photo-damage and photo-inhibition (Powles, 1984).

To compensate for fluctuations in light availability, marine phytoplankton have evolved extreme photo-physiological plasticity, allowing cells to maximize light harvesting capacity at low irradiance, while minimizing photo-damage under high

light levels. A better mechanistic understanding of the scope and limits of such coordinated regulation within the photosynthetic process is essential for the accurate modeling of 'bottom-up' controls on marine primary productivity and its response to environmental change. Furthermore, mechanistic insight into environmental controls on the light use efficiency of carbon fixation is crucial for the development of algorithms estimating primary productivity from remotely acquired optical data (Lee et al., 2015; Silsbe et al., 2016; Zoffoli et al., 2018).

In the present study, we examine diurnal variability in the capacity of phytoplankton to use light energy for biomass production in a productive coastal upwelling regime. High temporal resolution measurements, conducted over a 48 hour period, revealed coordinated changes in light absorption, energy dissipation, photosynthetic electron transport and $^{14}$C-uptake. Our results demonstrate strong variability in the stoichiometry of various components of the photosynthetic process, providing insight into phytoplankton metabolic acclimation potential in response to environmental fluctuation in coastal

waters. Comparison of these new results with previous observations in the iron-limited subarctic NE Pacific (Schuback et al., 2016), allowed us to identify distinct diurnal patterns in these contrasting environments, and yielded insight into the effects of iron limitation on various components of the photosynthetic process and their coupling over diurnal irradiance cycles. Most significantly, our data demonstrate a limited capacity of iron-limited phytoplankton to buffer fluctuations in light availability, resulting in an increased need for photo-protection. This enhanced photo-protection is achieved through



alterations in pigment ratios and light absorption characteristics, an increased potential for heat dissipation of excess energy (NPQ) and decoupling of the different components of the photosynthetic process, leading to reduced light use efficiency. Based on our results, we discuss the correlation between photosynthetic light use efficiency and NPQ, an optical signal amiable to high resolution acquisition by autonomous sensors.

5 ## 2 Methods

In the present study, we examined light-dependent diurnal variability in different components of the photosynthetic process in marine phytoplankton. We present new results from a 2017 research expedition in high productivity coastal upwelling waters, and compare these data to recently published observations from the iron-limited waters of the Subarctic Pacific Ocean (Schuback et al. 2016). We first introduce the two datasets, and then briefly describe the methods used to assess each 10 component of the photosynthetic process (Fig. 1), from light absorption to carbon fixation.

### 2.1 Dataset 1

New field data were collected during a 48 h period from August 19th to 21st, 2017 on board the R/V *Oceanus* in the subarctic NE Pacific. During this period, the research vessel followed a LaGrangian drifter equipped with a drogue sock at 5 m depth in order to track mean surface layer flow. The drifter was deployed approximately 25 nautical miles off the coast of Oregon, 15 USA (44.3 °N, 124.4 °W, Fig. 2). More information on the drifter study is available in Rosengard et al. (in prep).

Seawater samples were collected from the ship's underway water supply (intake depth approx. 5 m), and used for photo-physiological measurements by fast repetition rate fluorometry (FRRF; 2 h intervals), and $^{14}$C-uptake experiments, discrete sample collection for pigment analysis by HPLC and particulate light absorption (4 h intervals). Sample collection, handling and experimental protocols were identical to the methods used in Schuback et al. (2016). In the following, we provide only 20 brief details about sample analysis and rate measurements, with emphasis on approaches that extend beyond the analysis of Schuback et al. (2016). All measured variables and derived parameters are summarized in Table 1.

In addition to the discrete sample measurements described above, we acquired a number of additional datasets from various sensors connected to the ship's underway water supply. All measurements and sensors used on board the R/V *Oceanus* are summarised in Table S1.1 (OCE17 data set). Seawater surface temperature and salinity were measured by a 25 thermosalinograph (SBE 45 and SBE 38 for salinity and temperature, respectively), while surface PAR (400-700 nm) was continuously logged using a Satlantic PAR sensor mounted on the ship's superstructure. We used a Solience Fast Repetition Rate Fluorometer (FRRF) to continuously measure photo-physiological parameters derived from single turnover induction protocols (see section 3.5). In addition, we used WetLabs ac-s and BB3 sensors to quantify particulate light attenuation and absorption (400 – 750 nm), and particulate backscatter, following the protocols described in Burt et al. (2018).



## 2.2 Dataset 2

In a previous study (Schuback et al., 2016), we assessed variability and coupling of different components of the photosynthetic process in an iron-limited phytoplankton assemblage at Ocean Station Papa in the subarctic NE Pacific (50°N, 145 °W, Fig. 2). During this earlier study, conducted in June 2014, and hereafter referred to as OSP14, seawater

samples collected from the vessel's underway water supply (intake depth approx. 5 m) were used for photo-physiological measurements by FRRF (3 h intervals), [14]C-uptake experiments (3 h intervals), pigment analysis by HPLC (6 h intervals) and particulate light absorption (3 h intervals). All measurements taken are summarized in Table S1.2, and full details of sample handling, experimental protocols and instrumentation can be found in Schuback et al. (2016). In several instances, the dataset presented in Schuback et al. (2016) was reanalysed, as described below.

**2.3 Absorption spectra**

Phytoplankton absorption spectra ($a_{phy}(\lambda)$) were determined following the quantitative filter technique (QFT) of Mitchell et al. (2000) with path length amplification estimates following Bricaud and Stramski (1990), as described in detail in Schuback et al. (2017). All absorption spectra were corrected for an over-estimation of absorption at short wavelengths following the approach suggested by Letelier et al. (2017) and described in supplementary material S2. To determine

chlorophyll *a*-specific absorption spectra ($a^*_{phy}(\lambda)$, $m^2$ mg chl$a^{-1}$), absorption values were normalized to corresponding HPLC-derived [TChl*a*]. The chl*a*-specific phytoplankton absorption coefficient (400–700 nm) was calculated for a flat white spectrum ($â^*_{phy}$), and weighted to the spectrum of available light in situ ($ā^*_{phy}$) as described in Babin et al. (1996).

**2.4 Pigment analysis and spectral reconstruction**

Collection and analysis of HPLC pigment samples were performed following the method of Pinckney (2013), as described in

detail in Schuback et al. (2016). Pigment concentrations determined by HPLC and weight-specific absorption spectra provided by Bidigare et al. (1990) were used for reconstruction of phytoplankton light absorption spectra ($a^*_{phy}(\lambda)$). This approach estimates absorption spectra specific to photosynthetic pigments ($a^*_{psp}(\lambda)$) and photoprotective carotenoids ($a^*_{ppc}(\lambda)$). Following the approach described in Le et al. (2009) and Letelier et al. (2017), absorption spectra were further corrected for pigment packaging effects using a wavelengths-specific estimate of packaging developed by Morel and

Bricaud (1981), with a size parameter calculated from an empirical relationship to chlorophyll *a* concentration ([chl*a*]) (Woźniak et al. 1999). As described in the supplementary material (S2), we found good agreement between results from the spectral reconstruction and QFT approaches ($R^2 = 0.95$, n= 20).

**2.5 FRRF derived photophysiology**

Single-turnover induction curves of chl*a* fluorescence (ChlF) yields were measured on a bench-top FRRF instrument

(Soliense Instruments), after acclimation of samples to low light intensities (< 10 μmol quanta $m^{-2}$ $s^{-1}$) for 20 minutes. Blank



correction, derivation of ChlF yields and parameters, estimation of electron transport in reaction center II (ETR$_{RCII}$, mol e$^-$ mol RCII s$^{-1}$), and fitting of ETR$_{RCII}$ light response curves was performed as described in Schuback et al. (2016, 2017). We derived values of the maximum, light-saturated capacity of ETR$_{RCII}$ (ETR$_{RCII}$-P$_{max}$), the light-dependent increase of ETR$_{RCII}$ (ETR$_{RCII}$ -α), and rates for the in situ light intensity at the time and depth of sampling (Table 1).

We derived values of the minimum and maximum ChlF yields in the dark-regulated state (F$_o$, F$_m$), and in each light-regulated state of the light response curve (F', F$_m$'). The parameter F$_o$', which represents the minimum ChlF yield in the absence of photochemical quenching but presence of non-photochemical quenching, was estimated following Oxborough and Baker (1997). Chl$a$ fluorescence yields were used to estimate the ChlF parameter F$_v$/F$_m$ (= [F$_m$-F$_o$]/F$_m$), the maximum efficiency of absorbed light used for photochemistry, F$_q$'/F$_m$'(= [F$_m$'-F']/F$_m$'), the effective efficiency of absorbed light being

used for photochemistry, and F$_v$'/F$_q$'(= [F$_m$'-F$_o$']/[F$_m$'-F']), an estimate of the fraction of RCII in the 'open' state (Table 1). The functional absorption-cross section of RCII was derived in the dark-regulated (σ$_{PSII}$, Å RCII$^{-1}$) and light-regulated state (σ$_{PSII}$', Å RCII$^{-1}$), and spectrally-corrected to the spectral quality of in situ light (σ$_{PSII-IS}$), as described below. Non-photochemical quenching was estimated as normalized Stern-Volmer quenching, NPQ$_{NSV}$ (= F$_o$'/F$_v$') for each light level of the light response curves (McKew et al., 2013).

We note that the bio-physical model we used to derive photo-physiological parameters from FRRF measurements (Kolber and Falkowski, 1993; Kolber et al., 1998) is not likely to be equally accurate for all phytoplankton species within mixed in situ assemblages. Similarly, the fully dark-regulated state, necessary for the calculation of most ChlF parameters, is difficult to achieve in mixed assemblages consisting of species of varying NPQ mechanisms and capacities. As a result, the derived parameters represent best-guess average values for taxonomically-diverse phytoplankton assemblages.

**2.6 Photosynthetic unit size of PSII**

We estimated absolute values of the photosynthetic unit size of PSII (1/n$_{PSII}$, mol chl$a$ mol RCII$^{-1}$) following the approach suggested by Suggett et al. (2004). In this approach, 1/n$_{PSII}$ is obtained from FRRF-derived dark-regulated σ$_{PSII}$ (Å$^2$ RCII$^{-1}$), and photosynthetic pigment absorption spectra, a*$_{psp}$ (m$^2$ mg chl$a^{-1}$) estimated using the pigment reconstruction approach.

$$1/n_{PSII} = \frac{\sigma_{PSII}}{\bar{a}^*_{psp}} \cdot 0.013453 \qquad (1)$$

Here, both σ$_{PSII}$ and ā*$_{psp}$ are specific to the spectral distribution of the FRRF excitation LED. The factor 0.013453 converts mg chl$a$ to mol chl$a$, Å$^2$ to m$^2$, and RCII to mol RCII, and it is assumed that 50 % of absorbed photons go to PSII.

**2.7 $^{14}$C-uptake**

Rates of $^{14}$C-uptake were measured using small volume (20 ml), 2 hour light-response curves in a custom-built photosynthetron. Full details of the experimental procedure, calculation of rates, and fitting of light response curves can be

found in Schuback et al. (2016, 2017). As for light-response curves of ETR$_{RCII}$, we derived values of the maximum, light saturated capacity of $^{14}$C-uptake ($^{14}$C-P$_{max}$), and the light-dependent increase of $^{14}$C-uptake ($^{14}$C-α). From these two



parameters, we were able to derive $^{14}$C-uptake rates for the in situ light intensity at the time and depth of sampling (Table 1), using the exponential model of Webb et al. (1974).

Multiple studies have demonstrated that short-term $^{14}$C-uptake experiments, as employed here, measure an intermediate quantity between gross and net production (Halsey and Jones, 2015; Milligan et al., 2015; Pei and Laws, 2013). For fast growing, nutrient-replete phytoplankton (OCE17 in this study), a larger fraction of the initially fixed $^{14}$C will be rapidly respired, and the measured rate will therefore be closer to net productivity. For slow growing, nutrient-limited phytoplankton (OSP14 in this study) more of the initially fixed $^{14}$C will be retained in the cell, such that short incubation times will estimate rates closer to gross productivity. It is therefore likely that our derived $^{14}$C-uptake rates at OSP14 are over-estimated (closer to a gross rate) relative to OCE17 (closer to a net rate). This complicates the comparison of absolute $^{14}$C-uptake rates between the sites in the present study, but does not significantly change our conclusions regarding differences in the diel cycle of photosynthetic processes.

## 2.8 Spectral correction and derivation of stoichiometries

The spectral distribution of light at 5 m depth ($E_{is}(\lambda)$) was estimated as described in Schuback et al., (2016, 2017). Prior to curve fitting, absolute values of light intensity used for light response curves of $^{14}$C-uptake and ETR$_{RCII}$ ($E_{LED}(\lambda)$) were corrected relative to the phytoplankton light absorption spectrum.

$$E_{IS} = E_{LED} \cdot \frac{\sum_{400}^{700} a_{phy}(\lambda) E_{LED}(\lambda) \cdot \sum_{400}^{700} E_{IS}(\lambda)}{\sum_{400}^{700} a_{phy}(\lambda) E_{IS}(\lambda) \cdot \sum_{400}^{700} E_{LED}(\lambda)} \qquad (2)$$

Here, $a_{phy}(\lambda)$ is the phytoplankton absorption spectrum derived from the QFT approach. Values of $\sigma_{PSII}$, which are specific to the spectral distribution of excitation and background light in the FRRF instrument ($E_{LED}(\lambda)$), were corrected to the in situ spectral light distribution at time and depth of sampling ($E_{IS}(\lambda)$) using the same approach.

The electron requirement for carbon fixation ($\Phi_{e,C}$, mol e$^-$ mol C$^{-1}$, Fig. 1) was calculated by dividing chl$a$-specific rates of electron transport (ETR$_{RCII}$ / n$_{PSII}$) by chl$a$-specific rates of carbon fixation ($^{14}$C-uptake). The minimum value of $\Phi_{e,C}$, encountered during light limitation, was calculated using α values of each rate. The quantum efficiency of carbon fixation ($\Phi_C$, mol C mol photon absorbed $^{-1}$, Fig. 1) was calculated by dividing the rate of $^{14}$C-uptake by absorbed light ($\sum a_{phy}^* \cdot \sum PAR_{IS}$). The maximum photosynthetic efficiency, $\Phi_{C\text{-max}}$, was calculated as $\alpha^*$-$^{14}$C / $\bar{a}^*_{phy}$. Note that while $\Phi_C$ represents the quantum *efficiency* of carbon fixation (mol C mol photon$^{-1}$), $\Phi_{e,C}$ is generally is defined as the electron *requirement* of carbon fixation (mol e$^-$ mol C$^{-1}$).

## 3 Results and Discussion

In the following, we first describe the diurnal variability of the photosynthetic process during OCE17 experiment, from light absorption, via electron transport to carbon fixation (Fig. 1). We then compare the observed values and diurnal trends from this coastal upwelling regime to results obtained from a similar study in an iron-limited low biomass region (OSP14). Based





on this comparative analysis, we discuss the environmental controls on the regulation of the photosynthetic process, the magnitude and variability of the electron requirement and quantum efficiency of carbon fixation ($\Phi_{e,C}$ and $\Phi_C$, respectively), and the potential to use NQP measurements as a proxy for these important parameters.

### 3.1 Photosynthetic components and their diurnal periodicity during OCE17

Light absorption characteristics and PSII photo-physiology for the 48 hour diurnal cycle at OCE17 are summarized in Table 2. During our intensive sampling period, chl*a* biomass, derived from ac-s 676 nm absorption light height calibrated to HPLC [Tchl*a*], remained relatively constant ($1.08 \pm 0.15$ µg L$^{-1}$). Similarly, estimates of phytoplankton carbon biomass derived from continuous backscatter data with an empirical conversation factor (Graff et al., 2015) varied little during the 48 hour sampling period ($99.5 \pm 6$ µg C L$^{-1}$).

Derived values of $1/n_{PSII}$ ranged from 284 to 446 mol chl*a* mol RCII$^{-1}$, which is within the range of values measured in nutrient replete cultures and field assemblages using the oxygen-flash yield approach (e.g. Table 2 in Suggett et al., 2010). We observed no diurnal periodicity in the derived values of $1/n_{PSII}$, indicating that the number of functional RCII was not reduced by severe photo-damage during high mid-day irradiances (Table 2).

Phytoplankton absorption coefficients derived from QFT ($\hat{a}^*_{phy}$) ranged from 0.012 to 0.017 m$^2$ mg chl*a*$^{-1}$. Weighing these

estimates to the spectral distribution of in situ light ($\bar{a}^*_{phy}$) increased values by approximately 25%. No clear diurnal trend was observed in $\bar{a}^*_{phy}$.

The use of HPLC-derived absorption spectra allowed us to examine the contribution of photosynthetic and photo-protective pigments to total light absorption. The chl*a*-specific absorption coefficient of photosynthetic pigments ($\hat{a}^*_{psp}$) ranged from 0.009 to 0.011 m$^2$ mg chl*a*$^{-1}$, accounting for approximately 75 % of total phytoplankton absorption. By comparison, chl*a*-

specific absorption coefficients for photo-protective pigments, $\hat{a}^*_{ppc}$, were lower (approximately 25 % of total absorption), ranging from 0.0024 to 0.0046 m$^2$ mg chl*a*$^{-1}$. Both $\hat{a}^*_{psp}$ and $\hat{a}^*_{ppc}$ increased by approximately 20% when weighted to in situ light ($\bar{a}^*_{psp}$ and $\bar{a}^*_{ppc}$). We observed diurnal variability in the relatively contribution of these two pigment classes to total absorption, with the relative contribution of photo-protective carotenoids increasing during daylight hours (Fig. 3b).

In addition to the observed changes in pigment ratios, we observed a notable diel cycle in the functional absorption cross-

section, $\sigma_{PSII-IS}'$, and non-photochemical quenching, $NPQ_{NSV}$, derived for in situ light intensities (Table 2, Fig. 3c & 3d). Diurnal variability in these two parameters reflects regulation in the transfer of absorbed energy to RCII. The functional absorption cross-section exhibited a rapid decline following the onset of daylight, reaching minimum values at noon before increasing back to night-time maxima (Fig. 3c). $NPQ_{NSV}$ showed the opposite trend, with maximum values observed during mid-day, coincident with the minimum in $\sigma_{PSII-IS}'$ (Fig. 3d). The strong inverse correlation (Pearsons $\rho = 0.87$, $p < 0.001$, $n =$

22) between $\sigma_{PSII-IS}'$ and $NPQ_{NSV}$ is expected, and demonstrates that $NPQ_{NSV}$ is primarily attributable to thermal dissipation of excess excitation energy in the antenna (e.g. Xu et al., 2017).

Diurnal cycles in photo-protective pigment content and energy transfer within the pigment antenna (Fig. 3b-d) act to prevent excess excitation energy from reaching RCII, thus minimizing potential photo-damage (Fig. 1, process 1). Furthermore,



excitation energy at the level of RCII can be reduced by increasing the rate of charge separation and downstream electron transport (Fig. 1, process 2 and 3). Figure 3e shows the diel pattern in $F_q$'/$F_v$'(500), a variable which provides an estimate of the fraction of open RCII at a reference irradiance level of 500 µmol quanta m$^{-2}$ s$^{-1}$. Values of $F_q$'/$F_v$'(500) clearly followed the availability of light, indicating an increased ability to maximize the number of RCII in the open state ($Q_A$ oxidised) during high-light periods. The clear diurnal cycle in $F_q$'/$F_v$'(500) illustrates diurnal regulation of reactions downstream of light absorption and excitation energy transfer to RCII. This, in turn, implicates the up-regulation of reactions downstream of PSII (Fig. 1, process 3).

Parameters derived from $ETR_{RCII}$ and $^{14}$C-uptake light-response curves are shown in Table 3 and Fig. 4. $ETR_{RCII}$-$P_{max}$ ranged from 220 to 884 with a mean of 479 mol e$^-$ mol RCII$^{-1}$ s$^{-1}$. These values are in good agreement with values from previous studies (e.g. Hancke et al., 2015; Zhu et al., 2017), and fall below the theoretical maximum of 1000 mol e$^-$ mol RCII$^{-1}$ s$^{-1}$ for linear electron transport (Falkowski and Raven, 1997). Values of $ETR_{RCII}$-α ranged from 1.24 to 2.42, with a mean of 1.76 mol e$^-$ mol RCII$^{-1}$ s$^{-1}$ (µmol quanta m$^{-2}$ s$^{-1}$)$^{-1}$. The $E_k$ of $ETR_{RCII}$ varied from 160 to 410, with a mean of 262 µmol quanta m$^{-2}$ s$^{-1}$.

Clear diurnal periodicity in $P_{max}$, α, and $E_k$ of $ETR_{RCII}$ was observed in response to diurnal changes in light availability (Fig. 4c, e, g), with all three parameters showing maximum values during high irradiance mid-day periods. In situ light availability at the time and depth of sampling exceeded the $E_k$ for most of the day, meaning that $ETR_{RCII}$ at 5 m depth was not light-limited during a substantial portion of the day (Fig. 4a, g; note different scales on the panels).

Maximum rates of $^{14}$C-uptake ranged from 1.17 to 3.54 with a mean of 2.27 g C g chl$a^{-1}$ h$^{-1}$. Based on the high nutrient and biomass conditions at OCE17, we assume that phytoplankton growth rate was relatively high, such that these 2 h $^{14}$C-uptake experiments estimated a rate close to net productivity (e.g. Halsey and Jones, 2015; Milligan et al., 2015). Values of the light-dependent increase in $^{14}$C-uptake (α) ranged from 0.03 to 0.08 g C g chl$a^{-1}$ h$^{-1}$ (µmol quanta m$^{-2}$ s$^{-1}$), while the light-saturation parameter $E_k$ varied between 23 to 72 µmol quanta m$^{-2}$ s$^{-1}$ (Table 3, Fig. 4d, f, h). Clear diurnal trends were apparent in the $P_{max}$ of $^{14}$C-uptake (Fig. 4d), however, this trend was not observed for α, which decreased throughout each day (Fig. 4f). Values of $E_k$ of $ETR_{RCII}$ were always higher than $E_k$ of $^{14}$C-uptake, meaning that $^{14}$C-uptake saturated at light intensities at which $ETR_{RCII}$ remained light-dependent (Fig. 4c, d; note different scales on the panels).

An increase in the electron requirement for carbon fixation ($\Phi_{e,C}$, mol e$^-$ mol C$^{-1}$) is expected when $^{14}$C-uptake, but not ETR is light-saturated. Under such conditions, additional electrons from charge separation in RCII must be used for processes other than $^{14}$C-uptake (Fig. 1). As expected, values of $\Phi_{e,C}$ derived for in situ light availability (Table 3, Fig. 5b) showed a clear diurnal trend, closely following the diurnal change in light availability. Increased decoupling of $^{14}$C-uptake and $ETR_{RCII}$ under excess light (e.g. Corno et al., 2006; Fujiki et al., 2007; Schuback et al., 2017; Zhu et al., 2017) can be attributed to an upregulation of alternative electron sinks necessary to alleviate backpressure along the electron transport chain, once carbon fixation is saturated (e.g. Niyogi, 2000).

Figure 5 also shows diurnal trends in the quantum efficiency of carbon fixation, $\Phi_C$. This variable is influenced by the decoupling of electron transport and carbon fixation (i.e. $\Phi_{e,C}$, Fig. 1), and additionally by variations in the fraction of





absorbed light energy allocated to photochemistry (Fig. 1). Both the decoupling of electron transport from carbon fixation ($\Phi_{e,C}$, Fig 5a) and the quantum efficiency of carbon fixation ($\Phi_{C,}$ Fig. 5b) showed a clear dependence on diurnal variation in light availability.

## 3.2 Comparison between OCE17 and OSP14

The light-dependent photosynthetic response is strongly modified by environmental factors including temperature, nutrient availability, average light intensity, and light history (e.g. Sakshaug et al., 1997). Micro-nutrient limitation, most notably iron, has also been shown to exert a significant effect on light-dependent photosynthetic responses (Greene et al., 1991, 1992; Roncel et al., 2016; Schuback et al., 2015). Here, we examine potential iron-dependent effects by comparing absolute values and diurnal periodicity of components of the photosynthetic process between the high productivity coastal waters of

OCE17 and the iron-limited NE subarctic Pacific (OSP14, Schuback et al., 2016). Such a comparison is necessarily complicated by uncontrolled variability in a number of environmental and ecological factors, in addition to the iron status of resident phytoplankton assemblages. Nonetheless, we argue below that a clear signature of iron-limited physiology emerges from this comparison.

### 3.2.1 Comparison between environmental and ecological conditions between sampling sites

Table 4 summarizes hydrographic and biological properties of the two study sites. Temperature and salinity within the upper mixed layer were similar in both environments (11.5 °C and 32.6 PSU at OCE17, 10.4 °C and 32.4 PSU at OSP14), and the sites had well-defined mixed layers, with a depth of ~11 m at OCE17 and ~33 m at OSP14. Excess macronutrient concentrations were observed within the mixed layer of both stations (Table 4). However, micronutrients, most notably iron, were likely limiting phytoplankton growth at OSP14, thus accounting for the significantly lower [chl$a$] at this site (0.18 μg L$^{-1}$

, as compared to 1.04 μg L$^{-1}$ at OCE17, Table 4).

As expected, iron limitation also affected the phytoplankton community structure. We derived an estimate of phytoplankton community structure using pigment-based size classes (Claustre, 1994; Uitz et al., 2006; Vidussi et al., 2001). These estimates revealed that OCE17 was dominated by microphytoplakton (>20μm, ~67%), with ~33% of the phytoplankton assemblage attributable to the picophytoplankton size class (0.2 - 2μm). Based on the high concentration of the pigment

fucoxanthin, we assume that diatoms dominated the microphytoplankton size class in this region. Characteristic pigments for the nanophytoplankton size class (2-20μm, e.g. cryptophytes, chromophytes, and nanoflagellates) were present in very low concentrations at the OCE17 site, indicating a negligible contribution of this size to the phytoplankton assemblage. In contrast to the OCE17 site, the phytoplankton assemblage at OSP14 was dominated by picophytoplankton (~46%), with an estimated contribution of ~29% and ~25% for the nano and micro size classes, respectively. The high concentration of

zeaxanthin found at OSP14 suggests a high proportion of cyanobacteria in the smallest size class, while the relatively high values of 19'BF and 19'HF are characteristic for prymnesiohytes and pelagophytes. A summary of the HPLC pigment data is provide in S3.





Daylight hours at OSP14 were slightly longer than at OCE17 (~16 vs. 14 h, respectively), while daily integrated incident photon dose ($E_0$) was higher at OCE17 (36.21 vs. 31.94 mol quanta $m^{-2}$). However, given the greater water column light extinction coefficient ($k_d$, $m^{-1}$) at the OCE17 site (1.6 $m^{-1}$; vs. 0.7 $m^{-1}$), light availability calculated for the 5 m sampling depth was similar for the two sites (Table 4, Fig. 6d). In our analysis, we used instantaneous in situ light intensities to derive

photo-physiological parameters and [14]C-uptake rates from light-response curves. This approach is justified for a direct comparison of rates and diurnal patterns at a fixed depth. We note, however, that the deeper mixed layer at OSP14 likely affected the photo-acclimation status of the phytoplankton assemblage, as a result of stronger variability in light, and lower mean and median mixed layer irradiance levels.

### 3.2.2 Effects of iron limitation on photophysiology and diurnal regulation of photosynthesis

The photosynthetic electron transport chain has a high requirement for iron (Raven et al., 1999; Yruela, 2013) and iron limitation has been shown to exert a significant effect on the abundance and stoichiometry of its components (e.g. Davey and Geider, 2001; Ivanov et al., 2000; Strzepek and Harrison, 2004). Our data also clearly demonstrate this effect. The mean chl$a$-specific phytoplankton absorption coefficient, $\bar{a}^{*}_{phy}$ ($m^2$ mg chl$a^{-1}$), was 1.9 fold higher at OSP14 (Fig. 6a). This result can be explained by the smaller cell size and lower cellular [chl$a$] expected in iron-limited phytoplankton, both of which

reduce the packaging effect (Bricaud et al., 1995; Morel and Bricaud, 1981). We also observed a greater contribution of photo-protective pigments to light absorption at OSP14 (31%) relative to OCE17 (22%) (Fig. 6a). As discussed below, this result can be explained by the increased requirement for photo-protection under iron-limited growth conditions.

We found that the number of (iron-rich) PSII per chl$a$ ($n_{PSII}$, mol RCII mol chl$a^{-1}$) at OSP14, was approximately half of that observed at OCE17 (Fig. 6b). To partly compensate for this reduction in RCII, the dark-regulated functional absorption-cross

section ($\sigma_{PSII-IS}$, $Å^2$ $RCII^{-1}$) at OSP14 was almost 3 times higher than at OCE17 (Fig. 6c). This physiological response to iron limitation has been frequently observed in previous studies (Boyd et al., 2000; Kolber et al., 1994; Moore et al., 2007; Strzepek et al., 2012; Vassiliev et al., 1995).

Increased light absorption and charge separation per RCII observed at OSP14 creates the potential for over-saturation of the reaction centers, and resulting photo-inhibition. This, in turn, increases the requirement for active energy dissipation

mechanisms. Indeed, we observed strong diurnal adjustments in $\sigma_{PSII-IS}$' (Fig. 6e) and $NPQ_{NSV}$ (Fig. 6f), caused by active light-dependent regulation of excitation energy within the pigment antenna. Importantly, the dynamic range of light-regulated $\sigma_{PSII-IS}$' and $NPQ_{NSV}$ regulation over a diurnal cycle was significantly larger at OSP14 than at OCE17 (3-fold vs. 1.5-fold at OCE17), despite similar light intensities at the two sites (Fig. 6d-f). Our data therefore suggest an increased need for active regulation of energy dissipation in response to daily irradiance cycles in iron-limited waters.

Iron limitation comprises the plasticity of the photosynthetic process, and its ability to utilize high light intensities for carbon fixation. However, this does not lead to the reduction in light absorption, as one might expect of a system less capable to process light energy and more susceptible to damage by excess absorbed light. Rather, we observed an increased capacity for dissipation of excess absorbed energy through enhanced NPQ. Such a regulatory mechanism allows phytoplankton to





maximize photosynthesis under low light conditions, while preventing damage at high irradiances. Our results support previous observations showing high levels of NPQ in a variety of iron-limited phytoplankton in laboratory and field studies (e.g. Alderkamp et al., 2012; Allen et al., 2008; Hoppe et al., 2013; Petrou et al., 2014; Schallenberg et al., in prep.; Schuback et al., 2015; Terauchi et al., 2010; Vassiliev et al., 1995). A high NQP signature may thus hold potential as an

5 optical indicator for phytoplankton physiology and iron-nutrition status in the oceans (Schallenberg et al., in prep).

Further evidence of active regulation of excitation energy at the level of RCII can be seen in the high values and strong light-dependent increase in $ETR_{RCII}$ observed at OSP14 (Fig. 7b). The high mid-day rates of $ETR_{RCII}$ at OSP14 were not balanced by increased $^{14}C$-uptake (Fig. 7c), and exceeded the maximum theoretical value for linear electron transport. As described by Schuback et al. (2015, 2016), upregulation of alternative electron sinks, cyclic electron transport, and charge recombination

may all act to dissipate excess electrons and thereby prevent over-reduction of RCII. These mechanisms are manifested in an increase in $ETR_{RCII}$, and can account for the diurnal variation in the electron requirement of carbon fixation ($\Phi_{e,C}$, Fig 7c), with peak values observed in the mid-afternoon. Given that our $^{14}C$-uptake rates for OSP14 likely represent an upper bound (corresponding to GPP, as opposed to NPP for OCE17), the absolute values of $\Phi_{e,C}$ at OSP14 may be even higher than those presented in Fig. 7c.

As expected, values of $1/\Phi_C$ followed a pattern very similar to $\Phi_{e,C}$, with high values observed under super-saturating light intensities, and this light-dependent effect enhanced under iron limitation.

### 3.3 NPQ as optical signal

Our simultaneous measurements of light absorption, $ETR_{PSII}$, and $^{14}C$-uptake allowed us to calculate conversion factors between these rates, and observe variability in the electron requirement, $\Phi_{e,C}$ (mol e$^-$ mol C$^{-1}$), and quantum efficiency, $\Phi_C$

(mol C mol quanta absorbed$^{-1}$), of carbon fixation. Estimates of $\Phi_{e,C}$ are crucial to derive high spatial resolution carbon-based productivity estimates from FRRF measurements (e.g. Hughes et al., 2018b; Lawrenz et al., 2013), while the quantum efficiency of carbon fixation is a key parameter in absorption-based phytoplankton primary productivity models (Marra et al., 2007; Silsbe et al., 2016; Zoffoli et al., 2018). Determination of these parameters in the field is labor intensive, and it is therefore desirable to identify proxies that can be autonomously monitored at high resolution. Our results suggest that

estimates of NPQ, here derived from FRRF measurements, may provide useful information on both $\Phi_{e,C}$ and $\Phi_C$. NPQ is an optical signal amiable to high-resolution acquisition by autonomous sensors or remote sensing, which integrates the effects of multiple interacting environmental variables influencing photosynthetic energy conversion. As discussed in the following section, this parameter may hold unexploited potential to improve marine primary productivity estimates.

### 3.3.1 The electron requirement for carbon fixation, $\Phi_{e,C}$

Numerous studies have aimed to quantify variability in $\Phi_{e,C}$ in order to derive high resolution, FRRF-based estimates of phytoplankton productivity in carbon units (reviewed by e.g. Hughes et al., 2018b; Lawrenz et al., 2013). These studies have shown that $\Phi_{e,C}$ can vary widely, due to physiological regulation on short timescales, and taxonomic shifts on longer



temporal or larger spatial scales. In general, higher values of $\Phi_{e,C}$ are found under conditions of high excitation pressure at the level of RCII (high light and/or low nutrients). Indeed, for both OCE17 and OSP14, maximum $\Phi_{e,C}$ was observed during high irradiance periods in the afternoon, and $\Phi_{e,C}$, derived for in situ light availability followed PAR levels over the diurnal cycle (Fig. 7d). However, the diurnal range of $\Phi_{e,C}$ differed between OCE17 andOSP14, with a significantly larger range and

mid-day maximum in $\Phi_{e,C}$ in the iron-limited waters of OPS14 (Fig. 7d). This result suggests an enhanced need to dissipate excess electron pressure under iron-limiting conditions.

High excitation pressure also triggers the upregulation of heat dissipation mechanisms in the pigment antenna (here estimated as $NPQ_{NSV}$), and several studies have reported a correlation between $\Phi_{e,C}$ and $NPQ_{NSV}$ (Hughes et al., 2018a; Schuback et al., 2015, 2016b, 2017a; Zhu et al., 2017). We observed such a correlation at both of our sampling sites (Fig.

8a), but the slope of the $NPQ_{NSV} : \Phi_{e,C}$ correlation differed between the two sites (12.2 for OCE17 vs. 2.34 for OSP14; Fig. 8a). Several recent studies have similarly documented variability in the relationship between $\Phi_{e,C}$ and $NPQ_{NSV}$. For example, Hughes et al. (2018a) reported seasonally-dependent slopes between $NPQ_{NSV} : \Phi_{e,C}$ at a sampling site off the coast of Australia. In a previous study (Schuback et al., 2017), we observed a strong correlation between $NPQ_{NSV}$ and $\Phi_{e,C}/n_{PSII}$ in the upper mixed layer of the Arctic Ocean, but only very weak $NPQ_{NSV}$ and no apparent correlation with $\Phi_{e,C}/n_{PSII}$ below the

mixed layer.

Differences in experimental procedures and data analysis make it impossible to directly compare the slopes of $NPQ_{NSV}$ - $\Phi_{e,C}/n_{PSII}$ relationships between the different studies. Nonetheless, some general patterns do emerge. A strong correlation between $NPQ_{NSV}$ and $\Phi_{e,C}$ is likely to exist in all environments where phytoplankton must adapt to fluctuations in excitation pressure at the level of RCII. Such conditions result, for example, from high and fluctuating light intensities, nutrient

limitation and cold temperatures. However, the substantial taxonomic variability in phytoplankton photosynthetic architecture and photo-physiology (e.g. Campbell et al., 1998; Kunath et al., 2012) makes it likely that $NPQ_{NSV} : \Phi_{e,C}$ relationships will require regional tuning.

Variability in the $NPQ_{NSV} : \Phi_{e,C}$ relationship may limit the application of a single global approach to derive carbon-based productivity from FRRF data. Yet, such variability may hold inherent information about the physiological state of a

phytoplankton assemblage, and the bottom-up controls on primary productivity. For example, phytoplankton assemblages adapted to growth in high light and/or low nutrient environments appear to show stronger light-dependent increases in $NPQ_{NSV}$ than $\Phi_{e,C}$, leading to a change in the slope of the correlation between these variables. This result reflects preferential changes of the pigment antenna configurations (leading to heat dissipation as NPQ), over alterations of the electron transport chain and upregulation of alternative electron sinks (affecting $\Phi_{e,C}$). In this way, the slope of the $NPQ_{NSV} : \Phi_{e,C}$ correlation

may reflect taxonomic differences in evolutionary strategies to achieve balanced growth under varying environmental conditions.



### 3.3.2 The quantum efficiency of carbon fixation, $\Phi_C$

The quantum efficiency of carbon fixation, also referred to as photosynthetic efficiency, ($\Phi_C$) is defined as carbon fixed per unit of light absorbed. It is a fundamental biophysical parameter, which is poorly constrained in models of primary productivity (Hiscock et al., 2008; Silsbe et al., 2016; Sorensen and Siegel, 2001; Zoffoli et al., 2018). Regional variability in
its maximum value ($\Phi_{C-max}$), achieved under limiting light conditions, is evident in the comparison of our two study regions, with significantly lower values observed at OSP14 (0.038 ± 0.019 mol C mol photon $^{-1}$) relative to OCE17 (0.078 ± 0.019 mol C mol photon$^{-1}$). This result is consistent with laboratory and field observations showing a lower maximum quantum efficiency under nutrient limitation and / or low temperatures and high light environments (Babin et al., 1996b; Finenko et al., 2002; Marra et al., 2000; Morel, 1978; Ostrowska et al., 2012; Uitz et al., 2008).

The maximum quantum efficiency of carbon fixation is only achieved when photosynthesis is light-limited. At higher light intensities, absorbed light energy is increasingly redistributed to pathways other than $ETR_{RCII}$ (fluorescence or heat, Fig. 1, process 1), and excess energy within the electron transport chain is channeled to pathways other than carbon fixation (thereby increasing $\Phi_{e,C}$, Fig 1, processes 3 and 4). Consequently, values of $\Phi_C$ are expected to correlate with $NPQ_{NSV}$ (Fig. 8b) in a manner similar to that described for $\Phi_{e,C}$ (section 4.3.1). Indeed, NPQ has been extensively utilized as a proxy for
variability in $\Phi_C$ in remote sensing algorithms of terrestrial primary productivity (Gamon et al., 1997; Garbulsky et al., 2011; Peñuelas et al., 2013). In these terrestrial applications, light use efficiency (LUE, which is equivalent to $\Phi_C$) is estimated from the photochemical reflectance index (PRI), a proxy for NPQ derived from changes in reflectance band ratios, tracking changes in xanthophyll cycle (XC) pigments (e.g. Peñuelas et al., 2011). Relative to vascular plants, NPQ mechanisms in phytoplankton appear to be significantly more diverse, and are less well understood (e.g. Goss and Lepetit, 2015; Lavaud
and Goss, 2014). It is, therefore, unlikely that a PRI approach could be successfully applied to mixed phytoplankton assemblages across contrasting oceanic environments. We note, however, that light-dependent changes in pigment ratios have previously been correlated to $\Phi_C$ in phytoplankton (e.g Babin et al. 1996a; Johnson et al. 2002; Vaillancourt et al. 2003; Prieto et al. 2007; Marra et al., 2000). Moreover, changes in pigment ratios have been successfully correlated to absorption band ratios of phytoplankton (Eisner et al., 2003; Eisner and Cowles, 2005; Méléder et al., 2018; Stuart et al., 2000). These
results suggest that absorption band ratios may hold potential to improve estimates of $\Phi_C$ at regional scales.

A mechanistic coupling is expected between photo-protection and $\Phi_C$ across differing phytoplankton. Indeed, our results show that estimates of NPQ correlate well to measured values of $\Phi_C$ (Fig. 8b), suggesting that NPQ should be further examined as a proxy for $\Phi_C$. In the present study, we determined values of NPQ from FRRF measurements. However, other approaches to estimate NPQ in marine phytoplankton exist. These approaches have mostly been developed to correct for the
NPQ effect on phytoplankton chl$a$ biomass estimates from in situ ChLF sensors (Biermann et al., 2015; Thomalla et al., 2018; Xing et al., 2018), without fully exploiting the information inherent in this signal.

While previous studies have shown that irradiance provides an easily measurable proxy for changes in $\Phi_C$ (Kiefer and Mitchell, 1983; Silsbe et al., 2016) the response of $\Phi_C$ to incident light will be modulated by other environmental factors,




including nutrient availability and temperature. For example, the observed difference in $\Phi_{\text{C-max}}$ between OSP14 and OCE17 cannot be explained instantaneous light availability, which was similar at the two sites. In contrast, the regional difference we observed in $\Phi_{\text{C-max}}$ was well reflected in the extent of the diurnal NPQ response (Fig. 6f). NPQ thus provides an optical signal integrating a multitude of environmental controls on the photosynthetic apparatus, and may help constrain variability

in $\Phi_{\text{C}}$, leading to improved estimates of marine primary productivity.

## 4 Conclusion

The photosynthetic process plays a key role in the energy budget of phytoplankton metabolism, marine ecosystems, and the global carbon cycle. Yet, models of marine primary productivity typically do not adequately account for the dynamic environmental controls on photosynthesis and variations in the transfer of absorbed photon energy to organic carbon. The

present study aimed to enhance our understanding of the photo-physiological mechanisms maintaining energetic balance within the photosynthetic system of phytoplankton over diurnal timescales in contrasting marine environments.

Our results demonstrate how iron limitation affects the plasticity with which marine phytoplankton can optimize the use of light energy for carbon fixation. Low iron availability reduced the ability of phytoplankton to utilize diurnal increases in absorbed light energy for carbon fixation and increased the need for effective photo-protection.

Based on our data, we suggest that optical measurements of NPQ hold untapped potential to assess energy conversion efficiencies and, in turn, increase our ability to monitor phytoplankton physiology and primary productivity over a range of ecologically-relevant temporal and spatial scales.

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





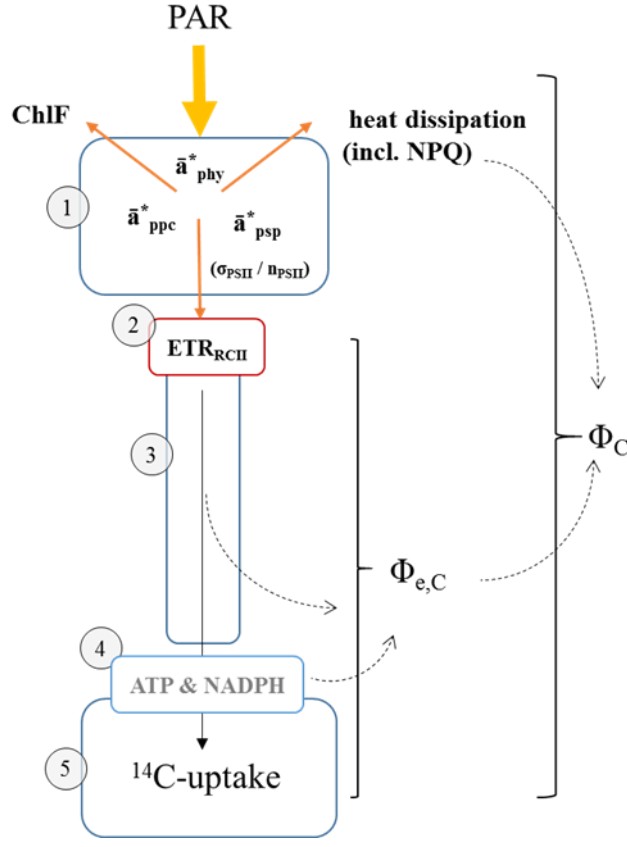

**Figure 1: Schematic diagram of the photosynthetic process, highlighting rates, variables, and conversion factors measured or derived during this study. 1 - Light absorption: Photosynthetically available radiation (PAR, 400 – 700 nm) is absorbed by phytoplankton ($\bar{a}^*_{phy}$, m$^{-2}$ mg chla$^{-1}$). Total absorption by phytoplankton can be subdivided into absorption by photosynthetic pigments ($\bar{a}^*_{psp}$, m$^{-2}$ mg chla$^{-1}$) and photoprotective carotenoids ($\bar{a}^*_{ppc}$, m$^{-2}$ mg chla$^{-1}$). The parameter $\bar{a}^*_{psp}$, if specific for PSII only, can be further decomposed into values of the functional absorption cross section of each RCII in the dark-regulated state, i.e. not affected by NPQ ($\sigma_{PSII}$, Å$^2$ RCII$^{-1}$) and the number of functional RCII per chla ($n_{PSII}$, RCII chla$^{-1}$). Both $\sigma_{PSII}$ and $n_{PSII}$ can be adjusted to regulate the amount of excitation energy reaching RCII. The light energy absorbed by the pigments of PSII can have three fates; photochemistry (ETR$_{RCII}$), dissipation as heat (including the upregulation of NPQ), and re-emission as fluorescence (ChlF). Changes in ChlF can be used to infer changes in the other two pathways. 2 - Initial charge separation in RCII (ETR$_{RCII}$, mol e$^-$ mol RCII$^{-1}$ s$^{-1}$). 3 - Electron transport after initial charge separation in RCII ultimately leads to the generation of 'photosynthate', (4. ATP and NADPH), which in turn can be used for carbon fixation (5. $^{14}$C-uptake). The electron requirement of carbon fixation $\Phi_{e,C}$ (mol e$^-$ mol RCII$^{-1}$) is the ratio of electrons displaced by initial charge separation in RCII to $^{14}$C-uptake. The photosynthetic efficiency, $\Phi_C$ (mol C mol quanta$^{-1}$), is the amount of $^{14}$C fixed per quanta absorbed. Under conditions when the rate of light absorption and delivery to RCII surpasses the potential for carbon fixation or reductant formation, both $\Phi_{e,C}$ and NPQ will increase to prevent over-reduction of RCII. The magnitude of $\Phi_C$, in turn, is dependent on how much initially absorbed energy is dissipated as fluorescence (ChlF) and heat (including NPQ) and through processes decoupling ETR$_{RCII}$ from $^{14}$C-uptake (reflected in $\Phi_{e,C}$).**





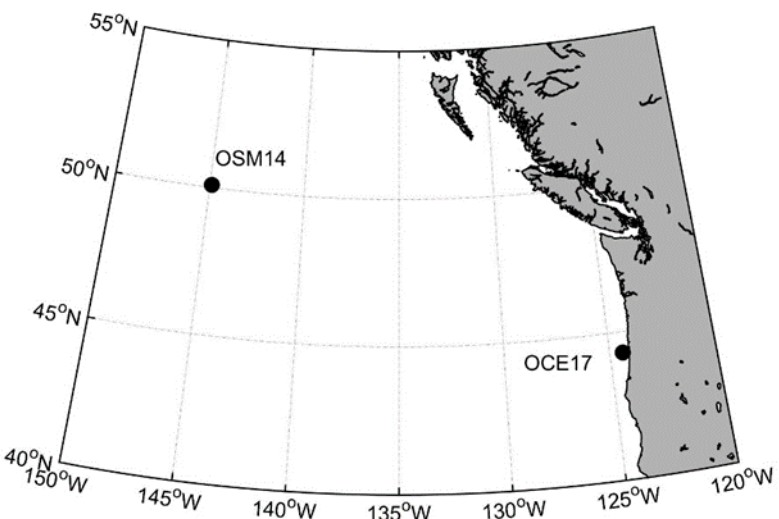

**Figure 2: Map of the NE subarctic Pacific, showing location of OSP14, in offshore iron-limited waters, and OCE17, a coastal upwelling region.**





**Table 1: List of parameters derived and discussed in the text.**

| Parameter | | Units | Method |
|---|---|---|---|
| $a^*_{phy\text{-}QFT}(\lambda)$ | Phytoplankton absorption spectra | $m^2$ mg chla$^{-1}$ | QFT with correction following Letelier et al. (2017). |
| $a^*_{xx\text{-}HPLC}(\lambda)$ | absorption spectra (xx specifies phytoplankton, photosynthetic pigments or photoprotective carotenoids) | $m^2$ mg chla$^{-1}$ | HPLC spectral reconstruction with packaging correction. |
| $\hat{a}^*_{xx}$ | Absorption coefficient | $m^2$ mg chla$^{-1}$ | Mean absorption 400 – 700 nm specific to flat white excitation light. |
| $\bar{a}^*_{xx}$ | Weighted absorption coefficient | $m^2$ mg chla$^{-1}$ | Mean absorption 400 – 700 nm weighted to in spectral distribution of situ light. |
| $\sigma_{PSII}$ | Functional absorption cross-section | $\text{Å}^2$ RCII$^{-1}$ | FRRF ST protocol during dark-regulated state, value specific to λ of excitation source. |
| $\sigma_{PSII\text{-}IS}$ | Functional absorption cross-section | $\text{Å}^2$ RCII$^{-1}$ | As above, value corrected to be specific to in situ light spectrum. |
| $\sigma_{PSII\text{-}IS}'$ | Functional absorption cross-section | $\text{Å}^2$ RCII$^{-1}$ | FRRF ST protocol during light-regulated state, value corrected to in situ light spectrum. |
| $F_v/F_m$ | Quantum efficiency of initial charge separation | no units | FRRF ST protocol during dark-regulated state; $(F_m - F_o)/F_m$. |
| $F_q'/F_v'$ (500) | Fraction of RCII which remain open ($Q_A$ oxidised) at a background irradiance of 500 µmol quanta $m^{-2}$ s$^{-1}$) | no units | FRRF ST protocol during light-regulated state; $(F_m' - F')/(F_m' - F_o')$. |
| $NPQ_{NSV}$ | Non-photochemical quenching at in situ light intensity at time and depth of sampling | no units | FRRF ST protocol during light-regulated state; $F_o'/(F_m' - F_o')$. |
| $NPQ_{NSV}$ (500) | Non-photochemical quenching for a reference light intensity of 500 µmol quanta $m^{-2}$ s$^{-1}$ | no units | As above. |
| $1/n_{PSII}$ | Photosynthetic unit size of PSII | mol chla mol RCII$^{-1}$ | Estimated from $\bar{a}^*_{psp}$ and $\sigma_{PSII-IS}$. |
| $ETR_{RCII}$ | Rate of initial charge separation in RCII | mol e$^-$ mol RCII$^{-1}$ s$^{-1}$ | Calculated from FRRF ST protocol derived parameters as $E \cdot \sigma_{PSII\text{-}IS}' \cdot F_q'/F_v'$. |
| $ETR_{RCII}$-$P_{max}$ | Maximum light saturated rate | mol e$^-$ mol RCII$^{-1}$ s$^{-1}$ | As above, but maximum rate of ETR achieved during light-response curve. |
| $ETR_{RCII}$-$\alpha$ | Light efficiency under light limitation | mol e$^-$ mol RCII$^{-1}$ s$^{-1}$ (µmol quanta $m^{-2}$ s$^{-1}$)$^{-1}$ | As above, but initial slope of light-response ETR curve. |
| $^{14}$C-uptake | Rate of carbon fixation | mol C mol chla$^{-1}$ s$^{-1}$ | 2 hr $^{14}$C-uptake light response curves measured at each time point. |
| $^{14}$C-$P_{max}$ | Maximum light saturated rate | mol C mol chla$^{-1}$ s$^{-1}$ | As above, but maximum rate of $^{14}$C-uptake achieved during light-response curve. |
| $^{14}$C-$\alpha$ | Light efficiency under light limitation | mol C mol chla$^{-1}$ s$^{-1}$ (µmol quanta $m^{-2}$ s$^{-1}$)$^{-1}$ | As above, but initial slope of $^{14}$C-uptake light response curve. |
| $E_k$ | Light saturation parameter | µmol quanta $m^{-2}$ s$^{-1}$ | Point of saturation during light response curve ($P_{max}/\alpha$) of ETR or $^{14}$C-uptake. |
| $\Phi_{e,C}$ | Electron requirement for carbon fixation | mol e$^-$ mol C$^{-1}$ | Calculated from ETR and $^{14}$C-uptake rates. |
| $\Phi_C$ | Quantum efficiency of carbon fixation | mol C mol quanta$^{-1}$ | Calculated from light absorption and $^{14}$C-uptake rates. |





**Table 2: Light absorption characteristics and PSII photo-physiology (process 1 in Fig. 1) for the 48 hour diurnal cycle at OCE17. Surface PAR (400-700 nm, µmol quanta m$^{-2}$ s$^{-1}$), during each sampling point. Total chlorophyll a (µg L$^{-1}$) from HPLC pigment analysis. Chlorophyll a-specific absorption coefficients for phytoplankton ($\bar{a}^*_{phy}$, m$^2$ mg chla$^{-1}$), and photosynthetic pigment ($\bar{a}^*_{psp}$, m$^2$ mg chla$^{-1}$), estimated using the HPLC pigment reconstruction approach and weighted to the spectral quality of in situ light.**

5     **The functional absorption cross-section of PSII, derived for the dark-regulated state ($\sigma_{PSII-IS}$, Å$^2$ RCII$^{-1}$) and specific to in situ light quantity at each sampling point ($\sigma_{PSII-IS}$', Å$^2$ RCII$^{-1}$), both corrected to the spectral quality of in situ light. Estimates of the photosynthetic unit size of PSII (1/np$_{PSII}$, mol chla mol RCII$^{-1}$). $F_v/F_m$, the maximum quantum efficiency of charge separation in RCII. $F_q'/F_v$' (500), an estimate of the fraction of 'open' reaction centers (Q$_A$ oxidised) at a reference irradiance of 500 µmol quanta m$^{-2}$ s$^{-1}$. NPQ$_{NSV}$, normalized Stern-Volmer quenching derived for in situ light intensity at time and depth of sampling.**

10     **NPQ$_{NSV}$, normalized Stern-Volmer quenching derived at a reference irradiance of 500 µmol quanta m$^{-2}$ s$^{-1}$. See methods section and table 1 for details on derivation of these parameters.**

| Local time | surface PAR | $\bar{a}^*_{phy}$ | $\bar{a}^*_{psp}$ | $\sigma_{PSII\text{-}IS}$ | $\sigma_{PSII\text{-}IS}$' | 1/np$_{PSII}$ | $F_v/F_m$ | $F_q'/F_v$' (500) | NPQ$_{NSV}$ | NPQ$_{NSV}$ (500) |
|---|---|---|---|---|---|---|---|---|---|---|
| 4:00 | 0 | 0.018 | 0.013 | 326 | 326 | 380 | 0.59 | 0.29 | 0.71 | 1.53 |
| 6:00 | 0 | | | 303 | 303 | | 0.57 | 0.41 | 0.76 | 1.60 |
| 8:00 | 175 | 0.017 | 0.014 | 300 | 263 | 313 | 0.57 | 0.45 | 0.82 | 1.74 |
| 10:00 | 188 | | | 317 | 279 | | 0.54 | 0.52 | 0.92 | 1.82 |
| 12:00 | 1054 | 0.019 | 0.01 | 325 | 223 | 446 | 0.49 | 0.57 | 1.5 | 1.89 |
| 14:00 | 1033 | | | 326 | 235 | | 0.43 | 0.64 | 1.89 | 2.47 |
| 16:00 | 1125 | 0.021 | 0.012 | 318 | 228 | 399 | 0.4 | 0.67 | 2.2 | 2.77 |
| 18:00 | 1163 | | | 316 | 221 | | 0.48 | 0.5 | 1.83 | 2.22 |
| 20:00 | 24 | 0.019 | 0.014 | 314 | 297 | 337 | 0.51 | 0.35 | 0.98 | 2.26 |
| 22:00 | 0 | | | 297 | 297 | | 0.52 | 0.3 | 0.93 | 2.39 |
| 0:00 | 0 | 0.018 | 0.013 | 303 | 303 | 338 | 0.52 | 0.3 | 0.93 | 2.36 |
| 2:00 | 0 | | | 306 | 306 | | 0.52 | 0.36 | 0.92 | 2.35 |
| 4:00 | 0 | 0.019 | 0.013 | 298 | 298 | 330 | 0.51 | 0.34 | 0.94 | 2.16 |
| 6:00 | 0 | | | 290 | 290 | | 0.5 | 0.39 | 1.02 | 2.29 |
| 8:00 | 270 | 0.017 | 0.013 | | | 338 | | | | |
| 10:00 | 1107 | | | 322 | 216 | | 0.47 | 0.67 | 1.73 | 2.19 |
| 12:00 | 1255 | 0.022 | 0.012 | 271 | 216 | 341 | 0.43 | 0.65 | 2.26 | 2.57 |
| 14:00 | 1431 | | | 271 | 195 | | 0.35 | 0.73 | 2.63 | 2.87 |
| 16:00 | 1085 | 0.019 | 0.013 | 302 | 221 | 348 | 0.38 | 0.73 | 2.18 | 2.87 |
| 18:00 | 347 | | | 314 | 292 | | 0.45 | 0.58 | 1.56 | 2.69 |
| 20:00 | 24 | 0.015 | 0.011 | 272 | 270 | 382 | 0.41 | 0.48 | 1.55 | 4.23 |
| 22:00 | 0 | | | | | | | | | |
| 0:00 | 0 | 0.015 | 0.014 | 268 | 268 | 284 | 0.54 | 0.27 | 0.86 | 2.06 |
| 2:00 | 0 | | | 277 | 277 | | 0.5 | 0.35 | 0.99 | 2.68 |





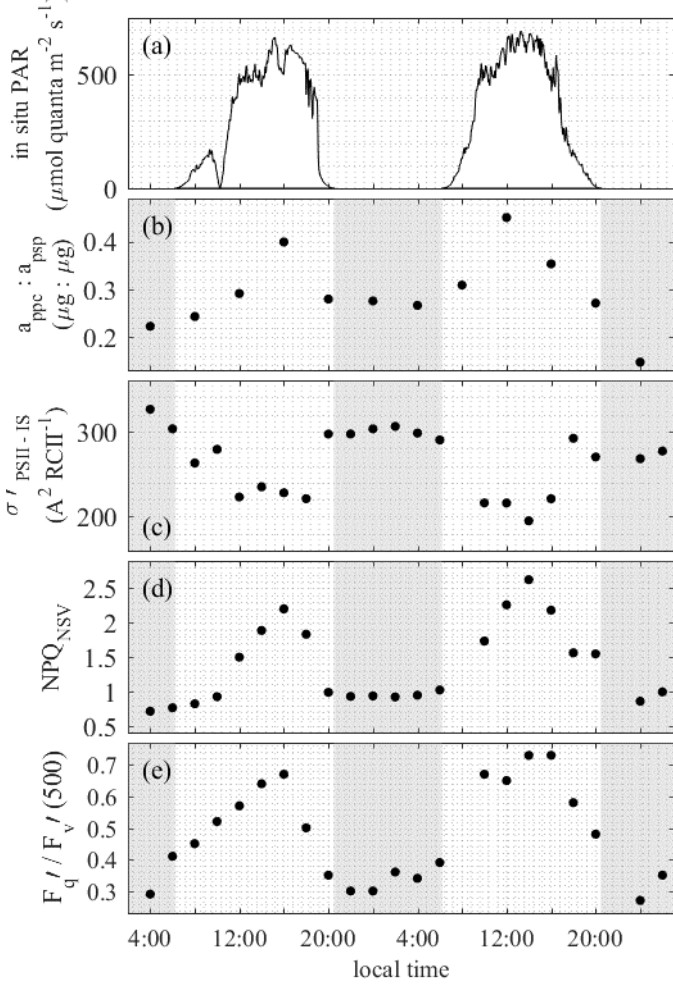

**Figure 3: Diurnal variability in light absorption and energy transfer in the light-harvesting antenna of PSII (Fig. 1, process 1) at the OCE17 site. (a) PAR estimated for 5 m sampling depth. (b) Ratio of absorption by photoprotective carotenoids ($\bar{a}^*_{ppc}$) to absorption by photosynthetic pigment ($\bar{a}^*_{psp}$), where both values are derived from spectral reconstruction of HPLC pigment data. (b) Values of $\sigma'_{PSII-IS}$, spectrally corrected to in situ spectral light quality, derived from FRRF light response curves at light levels corresponding to in situ light intensity. (c) Values of $NPQ_{NSV}$ derived from FRRF light response curves at light levels corresponding to in situ light intensity at the time and depth of sampling. (d) Values of $F_q'/F_v'$ derived from FRRF light response curves at a reference background irradiance of 500 µmol quanta m$^{-2}$ s$^{-1}$.**



**Table 3: Light-response curve fit parameters for rates of charge separation in RCII (ETR$_{RCII}$) and $^{14}$C-uptake for the 48 hour diurnal cycle at OCE17. Units of ETR$_{RCII}$ are mol e$^-$ mol RCII$^{-1}$ s$^{-1}$ and units of $^{14}$C-uptake are g C g chla$^{-1}$ h$^{-1}$. P$_{max}$ is the maximum rate at light saturation, α is light efficiency of each rate under light limitation, and E$_k$ is the light saturation parameter (µmol quanta m$^{-2}$ s$^{-1}$). The errors given are the 95% confidence interval for the fit parameter P$_{max}$ and α, and the propagated error for E$_k$. In situ (IS) are realized rates derived for in situ light intensities for the time and depth of sampling. Φ$_{e,C}$ is the electron requirement for carbon fixation (mol e$^-$ mol C$^{-1}$) and Φ$_C$ the quantum efficiency of carbon fixation (mol C mol photon absorbed$^{-1}$). The minimum value of Φ$_{e,C}$ and maximum value of Φ$_C$ at each time-point are theoretical values describing the acclimation state of the entire photosynthetic process. In situ (IS) values are realized values of Φ$_{e,C}$ and Φ$_C$ derived for in situ light intensities at the time and depth of sampling.**

| local time | ETR$_{RCII}$ P$_{max}$ | α | E$_k$ | IS | $^{14}$C-uptake P$_{max}$ | α | E$_k$ | IS | Φ$_{e,C}$ min | IS | Φ$_C$ max | IS |
|---|---|---|---|---|---|---|---|---|---|---|---|---|
| 4:00 | 295 ± 8 | 1.49 ± 0.01 | 198 ± 13 | 0 | 1.7 ± 1.1 | 0.06 ± 0.14 | 29 ± 70 | 0 | 3 ± 8 | 0 | 0.08 | |
| 6:00 | 412 ± 13 | 1.62 ± 0.01 | 253 ± 16 | 0 | | | | | | | | |
| 8:00 | 426 ± 17 | 1.79 ± 0.14 | 238 ± 21 | 120 | 2.2 ± 0.6 | 0.08 ± 0.09 | 27 ± 31 | 2 | 4 ± 4 | 9 | 0.11 | 0.03 |
| 10:00 | 532 ± 12 | 1.94 ± 0.07 | 274 ± 12 | 142 | | | | | | | | |
| 12:00 | 623 ± 20 | 2.26 ± 0.13 | 276 ± 18 | 512 | 3.0 ± 0.3 | 0.08 ± 0.03 | 39 ± 16 | 3 | 3 ± 1 | 19 | 0.09 | 0.01 |
| 14:00 | 725 ± 15 | 2.42 ± 0.08 | 299 ± 11 | 572 | | | | | | | | |
| 16:00 | 729 ± 16 | 2.35 ± 0.08 | 311 ± 12 | 585 | 3.5 ± 0.4 | 0.06 ± 0.02 | 62 ± 20 | 3.5 | 5 ± 1 | 20 | 0.06 | 0.01 |
| 18:00 | 489 ± 12 | 2.07 ± 0.11 | 237 ± 13 | 435 | | | | | | | | |
| 20:00 | 340 ± 10 | 1.66 ± 0.09 | 204 ± 14 | 18 | 2.5 ± 0.3 | 0.06 ± 0.03 | 38 ± 18 | 0.6 | 4 ± 2 | 4 | 0.08 | 0.07 |
| 22:00 | 237 ± 13 | 1.47 ± 0.19 | 162 ± 24 | 0 | | | | | | | | |
| 0:00 | 261 ± 13 | 1.29 ± 0.14 | 203 ± 24 | 0 | 1.2 ± 0.1 | 0.04 ± 0.02 | 29 ± 13 | 0 | 5 ± 2 | 0 | 0.05 | |
| 2:00 | 331 ± 23 | 1.24 ± 0.15 | 266 ± 38 | 0 | | | | | | | | |
| 4:00 | 321 ± 13 | 1.37 ± 0.11 | 235 ± 20 | 0 | 1.7 ± 0.2 | 0.08 ± 0.03 | 23 ± 10 | 0 | 3 ± 1 | 0 | 0.09 | |
| 6:00 | 345 ± 29 | 1.48 ± 0.24 | 234 ± 42 | 0 | | | | | | | | |
| 8:00 | | | | | 2.0 ± 0.4 | 0.06 ± 0.05 | 36 ± 29 | 1.9 | | | 0.08 | 0.02 |
| 10:00 | 807 ± 35 | 2.14 ± 0.12 | 378 ± 27 | 591 | | | | | | | | |
| 12:00 | 602 ± 21 | 1.75 ± 0.09 | 344 ± 21 | 485 | 3.2 ± 0.4 | 0.06 ± 0.03 | 51 ± 23 | 3.2 | 4 ± 2 | 21 | 0.07 | 0.01 |
| 14:00 | 784 ± 23 | 2.14 ± 0.09 | 366 ± 18 | 649 | | | | | | | | |
| 16:00 | 884 ± 42 | 2.16 ± 0.13 | 410 ± 31 | 615 | 3.0 ± 0.3 | 0.04 ± 0.01 | 68 ± 20 | 3 | 7 ± 2 | 28 | 0.05 | 0.01 |
| 18:00 | 579 ± 14 | 2.02 ± 0.08 | 286 ± 13 | 243 | | | | | | | | |
| 20:00 | 305 ± 15 | 1.39 ± 0.14 | 219 ± 24 | 15 | 2.0 ± 0.2 | 0.03 ± 0.01 | 72 ± 22 | 0.3 | 6 ± 2 | 7 | 0.04 | 0.03 |
| 22:00 | | | | | | | | | | | | |
| 0:00 | 220 ± 8 | 1.37 ± 0.12 | 160 ± 15 | 0 | 1.2 ± 0.6 | 0.05 ± 0.08 | 27 ± 52 | 0 | 5 ± 9 | 0 | 0.07 | |
| 2:00 | 284 ± 21 | 1.30 ± 0.19 | 220 ± 36 | 0 | | | | | | | | |





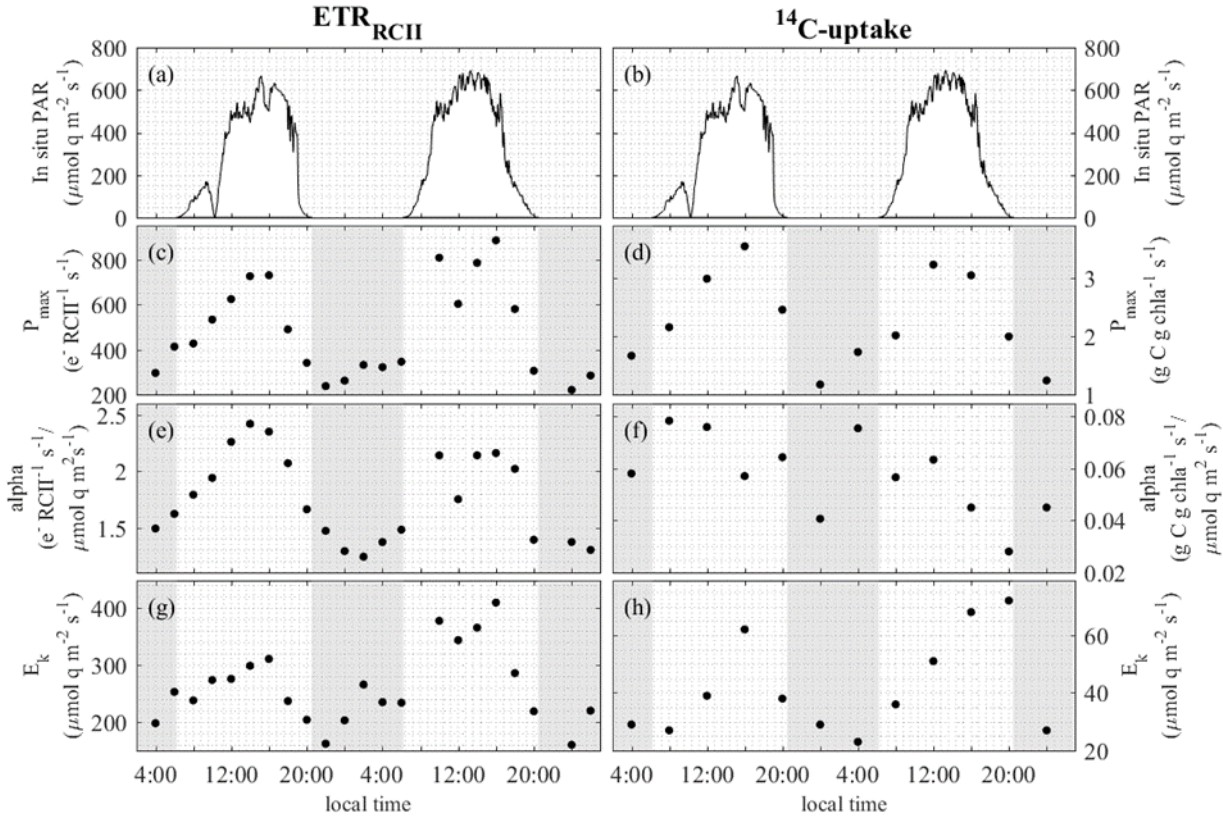

**Figure 4: Diurnal variability in light response curve fit parameters for ETR$_{RCII}$ (left) and $^{14}$C-uptake (right) for the OCE17 site. Panels (a) and (b) show PAR at 5 m sampling depths. Panels (c) and (d) show the maximum, light saturated capacity P$_{max}$ of each rate. Panels (e) and (f) show the light efficiency of each rate under light limitation, α. Panels (g) and (h) show the light saturation parameter E$_k$ of each rate. Note different scales on (a), (b), (g), and (h).**





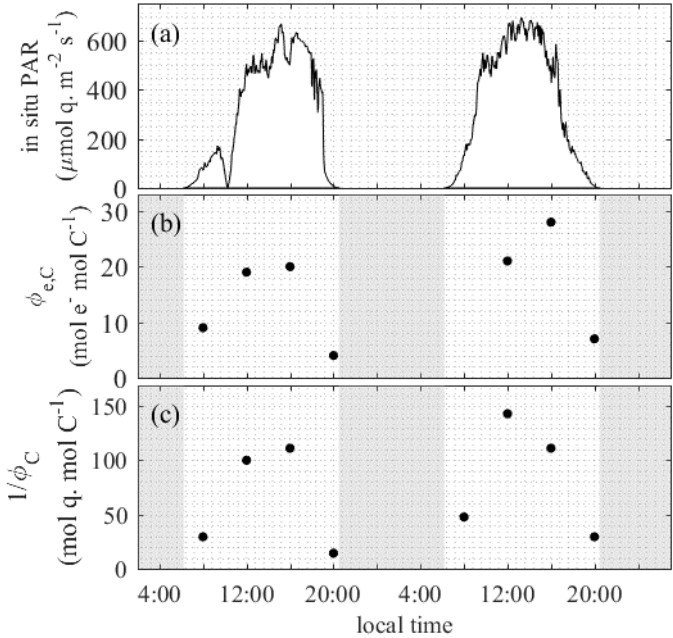

**Figure 5: The realized electron and photon requirements of carbon fixation over a 48 hour diurnal cycle at the OCE17 site. Values of Φe,C and 1/ΦC correspond to light conditions at time and depth of sampling. Note that we present the photon requirement for carbon fixation, 1/ΦC, instead of the photon efficiency of carbon fixation (ΦC), to facilitate better comparability with Φe,C.**





**Table 4: Comparison of environmental conditions at the OSP14 and OCE17 sampling sites. See text for details on derivation of each variable.**

| | | OSP14 | OCE17 |
|---|---|---|---|
| **Date** | *DD.MM.YY* | 17.06.14 | 21.08.17 |
| **Lat** | *° N* | 50.1 | 144.9 |
| **Long** | *° W* | 44.3 | 124.4 |
| **sunrise (PDT)** | *hh:mm* | 6:27 | 6:26 |
| **sunset (PDT)** | *hh:mm* | 22:49 | 20:12 |
| **daylength** | *hh:mm* | 16:22 | 13:46 |
| **$k_d$** | *$m^{-1}$* | 0.07 | 0.16 |
| **$E_0$** | *mol quanta $m^{-2}$ $d^{-1}$* | 31.94 | 36.21 |
| **$E_{ml}$ mean** | *mol quanta $m^{-2}$ $d^{-1}$* | 12.45 | 17.03 |
| **$E_{ml}$ median** | *mol quanta $m^{-2}$ $d^{-2}$* | 10.1 | 15.0 |
| **$E_{5m}$** | *mol quanta $m^{-2}$ $d^{-1}$* | 21.4 | 16.24 |
| **$E_{5m}$ mean** | *µmol quanta $m^{-2}$ $s^{-1}$* | 281 | 313 |
| **$E_{5m}$ max** | *µmol quanta $m^{-2}$ $s^{-1}$* | 802 | 661 |
| **temp** | *°C* | 10.4 | 11.5 |
| **salinity** | *PSU* | 32.4 | 32.6 |
| **MLD** | *m* | 33 | 11 |
| **$[NO_3+NO_2]$** | *µM* | 9.1 | 8.6 |
| **[P]** | *µM* | 0.98 | 0.8 |
| **[Si]** | *µM* | 14.5 | 9.8 |
| **[Tchla]** | *µg $L^{-1}$* | 0.18 | 1.04 |
| **f_micro** | *%* | 67 | 25 |
| **f_nano** | *%* | 0 | 29 |
| **f_pico** | *%* | 33 | 46 |



**Figure 6: Comparison of light absorption characteristics at the OSP14 and OCE17 sampling sites. Panel (a): the mean (400-700 nm) chla-specific absorption coefficient of phytoplankton (â\*$_{phy}$, m$^2$ mg chla$^{-1}$), showing contribution of absorption by photosynthetic pigment (â\*$_{psp}$) and photoprotective carotenoids (â\*ppc). Panel (b): photosynthetic unit size of PSII, 1/n$_{PSII}$ (mol chla mol RCII$^{-1}$). Panel (c): the functional absorption cross section of PSII, σ$_{PSII}$ (Å$^2$ RCII$^{-1}$), derived for the dark-regulated state at each time-point. In (b) and (c) the central mark in each box is the median, the edges of the box are the 25th and 75th percentiles, and the whiskers extend to the range of all data. No clear diurnal trend in 1/n$_{PSII}$ or σ$_{PSII}$ was detected at either station. Panel (d): values of PAR (400-700 nm, μmol quanta m$^{-2}$ s$^{-1}$) at 5 m sampling depth. Panel (e): the functional absorption cross section of PSII, σ'$_{PSII}$ (Å$^2$ RCII$^{-1}$), measured at the light-regulated state corresponding to in situ light intensity at each time-point. Panel (f): non-photochemical quenching, measured at the light-regulated state corresponding to in situ light intensity at each time-point.**





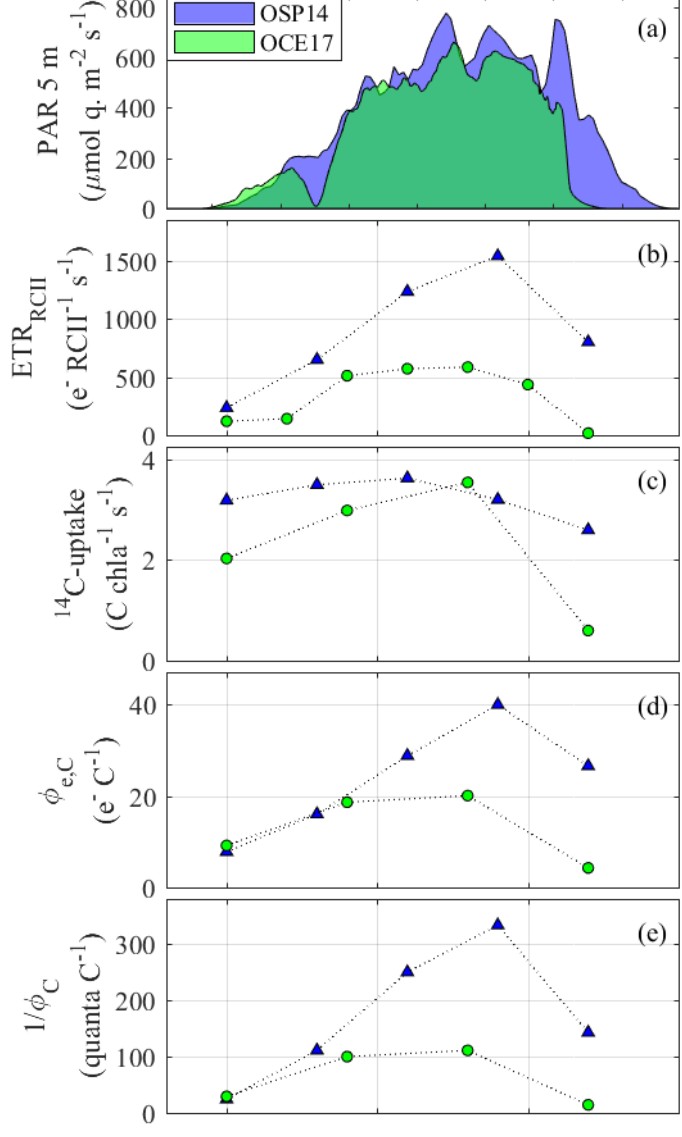

**Figure 7: Comparison of diurnal trends observed at the OSP14 and OCE17 sampling sites. Panel (a): PAR estimated for 5 m depth. Panel (b): rates of initial charge separation in individual RCII (ETR$_{RCII}$). Panel (c): rates of $^{14}$C-uptake. Panel (d): the electron requirement for carbon fixation ($\Phi_{e,C}$). Panel (e): the quantum requirement for carbon fixation (1/$\Phi_C$). All rates and efficiencies correspond to in situ light availability at time and depth of sampling.**





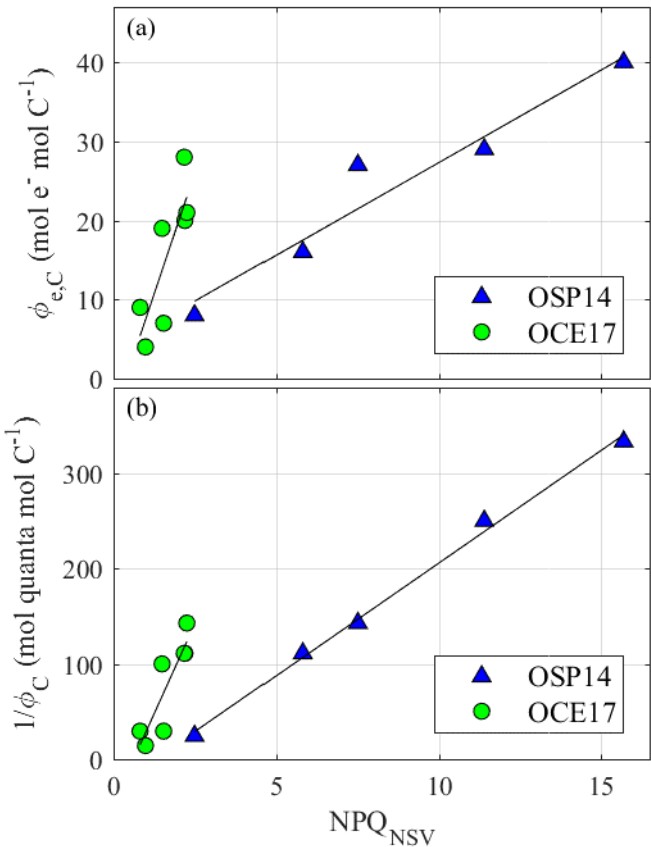

**Figure 8: (a) Correlations between the electron requirement of carbon fixation, $\Phi_{e,C}$ and NPQ$_{NSV}$, both derived for in situ light intensity at time and depth of sampling for each time-point during daylight hours. OCE17: $\Phi_{e,C}$ = 12.2 * NPQ$_{NSV}$ - 4.53; $R^2$ = 0.68; n=7. OSP14: $\Phi_{e,C}$ = 2.34 * NPQ$_{NSV}$ + 3.74; $R^2$ = 0.94, n=5. (b) Correlations between the photon requirement of carbon fixation,**
5 **1/$\Phi_C$ and NPQ$_{NSV}$, both derived for in situ light intensity at time and depth of sampling for each time-point during daylight hours. OCE17: 1/$\Phi_{,C}$ = 75 * NPQ$_{NSV}$ - 46; $R^2$ = 0.68; n=7. OSP14: 1/$\Phi_C$ = 23 * NPQ$_{NSV}$ + 30; $R^2$ = 0.94; n=5.**