# Peer review of "Diurnal regulation of photosynthetic light absorption, electron transport and carbon fixation in two contrasting oceanic environments"

_Biogeosciences, 2018_

## Referee Comment (RC1) · Anonymous Referee #1 · 10 Feb 2019

general commentsïijŽ The manuscript by Schuback and Tortell examined the variability of several parameters including phytoplankton absorption, FRRF-ETR and primary productivity over 48 hours in the coastal subarctic NE pacific, which I believe should be a very hard work. Moreover, they also compared results of this study with their previous one from an iron limited area, to give the idea that the potential effects of iron limitation on photosynthesis. They showed the first time that NPQ is a good factor for estimating both $\Phi e,C$ and $\Phi C$, which could contribute to FRRF, numerical models and remote sensing based primary production estimates. It seems that the authors' data

set, results and supplement files are very well prepared and is comprehensive enough to address these aspects of light absorption, electron transport and carbon fixation diurnal regulation. So I do not notice any major concerns with the manuscript, but have some concerns and question want to ask/suggest.

specific commentsïijŽ Page 1, line 4: what is NE Pacific? Page 1 line 19-20: the meaning of this sentence is not very clear for me, so author is saying, comparing to the coastal waters, although under iron-limitation there was a significant reduction of iron-rich photosynthetic units per chlorophyll a (this I can understand), the electron transport per photosystem II is still higher. ( Is it right?) If so what cause this higher ETR? Page 1, line 20: put PSII after photosystem II ?, because PSII will soon show up in the caption of Fig.1 Page 2, line 11: I think Fig.1 is a very nice schematic plot, just two suggestions 1) Make difference for a*phy and a*ppc, a*psp, now it seems these three parameters are equal ., 2) I think not very necessary to put 14C here, just C-uptake is OK. Page 2, line 12: maybe add some references here? Page 5ïijŇline 25: not very clear why here a ÌĔ*psp is weighted to FRRF excitation LED, not in situ light? Page 5ïijŇline 26: using assumption that ratio of PSII: PSI =1 whether will affect the accuracy of nPSII calcuation? especially for those samples under iron limitation, which should have decreased PSI abundance. Can authors provide the general range of PSII:PSI for samples with/without iron stress? Page 6ïijŇ line16ïijŽI feel eq.2 is very hard to follow, here are some questions 1) where is the $Eis(\lambda)$ in Eq.(2) ? 2) Maybe I missed somewhere but I cannot find where you mention that $Eis(\lambda)$ (i.e. Eis at each wavelength) was measured? or you measured $E0+(\lambda)$ ?, then using it to estimate $Eis(\lambda)$. Sorry, just cannot find the related information. 3) not very clear why absolute values of light intensity for 14C P-E curve need to be corrected? And how can you correct light? I think you can only correct 14C-uptake rates, because C uptake rate measured under indoor LED light may differ with that under in situ natural light Page 6ïijŇ line24: a little confused that why $\Phi c\text{-}max = \alpha^*\text{-}14C / ÄĄ^*phy$? is not $=Pmax\text{-}14C / ÄĄ^*phy$ ? Page 6, line20-25: I would suggest authors adding equations for how to calculate $\Phi e,C$, and $\Phi C$ here. For me it is not very easy to get $\Phi e,C$ because the unit

of ETR per second, but C-uptake is per hour. And it is same to ΦC. I think it will help to understand the meaning of Φe,C ΦC if authors can provide equations and parameters with unit here Page 7, line8-9: I think the datasets Graff (2015) used for developing their bbp-Cphyto algorism mostly came from Open Ocean, where the phytoplankton is the main particle; however, when it is not the case (usually refer to Case II water), I think the algorism may not be suitable here, unless in this study area the backscatter signal mainly come from phytoplankton. And also, the author didn't provide the description of how they correct the backscatter data, so I would suggest authors to remove the phytoplankton carbon part. Page 7, line 25, table 2: previously I thought NPQ should be highly correlated with surface PAR, but actually from the results in table 2 we can found oblivious "decoupling" exists within these two parameters. For example the second 24 hours 20:00, when the PAR is only 24, the NPQ value is actually higher than the NPQ at first 24 hours 12:00, when the PAR is 1054, do authors know the reason? Page 11, line 8: can you explain the reason of what may potentially cause mid-day ETR at OSP14 exceeds the maximum theoretical value Page 11, line 8: as authors mentioned, the weak part of this MS is figure 7c, which is not very easy for primary productivity people to understand. It is telling that at OSP14, even water dominated by smaller phytoplankton and has nutrient limitation; it still has higher PB, which I think against most of the primary production research. Although it might be explained by NPP/GPP reason, I suggest in the future study authors should try to give or adjust the primary productivity rate to same level, for example, also measure respiration rate at same time. Page 11, line 26:-27 adding some references here?

technical correctionsïijŽ Page 2 line 25: "we examined diurnal variability……." Page 3, line 29, I cannot find Burt et al. (2018) in references Page 7, line3 and Page 11, line 4ïijŽtyping errors, "NQP" should be "NPQ" Page 11, line 11 and 14, should be Fig 7d Page 11, line 15, Fig 7e is missing here Page 23, Figure 2: typing errors, "OSM14"in figure should be "OSP14" Page 24, Table 1: method column, fourth item, "……weighted to spectral distribution of in situ light" Page 32 Figure 7: missing x-axis label Page 28, 32, Figure 4 and 7. The unit for 14C-uptake should be per hour, not per

second

---

## Author Comment (AC1) · 15 Feb 2019

Reply to anonymous referee #1, in blue.

General comments:

The manuscript by Schuback and Tortell examined the variability of several parameters including phytoplankton absorption, FRRF-ETR and primary productivity over 48 hours in the coastal subarctic NE pacific, which I believe should be a very hard work. Moreover, they also compared results of this study with their previous one from an iron limited area, to give the idea that the potential effects of iron limitation on photosynthesis. They showed the first time that NPQ is a good factor for estimating both $\Phi e,C$ and $\Phi C$, which could contribute to FRRF, numerical models and remote sensing based primary production estimates. It seems that the authors' data set, results and supplement files are very well prepared and is comprehensive enough to address these aspects of light absorption, electron transport and carbon fixation diurnal regulation.

So I do not notice any major concerns with the manuscript, but have some concerns and question want to ask/suggest.

We thank the referee for their kind words and answer all specific comments below.

Specific comments:

Page 1, line 4: what is NE Pacific?

We changed the text to "North-East Subarctic Pacific".

Page 1 line 19-20: the meaning of this sentence is not very clear for me, so author is saying, comparing to the coastal waters, although under iron-limitation there was a significant reduction of iron-rich photosynthetic units per chlorophyll a (this I can understand), the electron transport per photosystem II is still higher. (Is it right?) If so what cause this higher ETR?

The referees understanding is correct, and we appreciate that higher ETR under iron limitation appears counter-intuitive. As explained in much detail throughout the manuscript:

- Under iron-limited conditions, more chlorophyll a is associated with each (iron-rich) photosystem II. This allows cells to reduce cellular iron demand by increasing light absorption and energy transport to each reaction center II (RCII).
- As a direct consequence, the rate of charge separation per RCII ($ETR_{RCII}$) increases under iron limitation. It can be visualized as 'funneling' more electrons through the same RCII in order to reduce cellular iron demand.

The important point is that if ETR was normalized to [chla], the rate would of course be lower in the iron-limited region.

We revised the manuscript to make this point clearer in the abstract, and note that it is explained in much detail within the manuscript.

Page 1, line 20: put PSII after photosystem II?, because PSII will soon show up in the caption of Fig.1

Done.

Page 2, line 11: I think Fig.1 is a very nice schematic plot, just two suggestions 1) Make difference for a*phy and a*ppc, a*psp, now it seems these three parameters are equal., 2) I think not very necessary to put 14C here, just C-uptake is OK.

The distinction between a*phy and a*ppc and a*psp was described in the original figure legend. We have further revised the figure to read a*phy= a*ppc + a*psp. "14C-uptake" was replaced by "C-fixation", as suggested. We have now indicated in the revised legend that C-fixation was estimated by $^{14}$C-uptake.

Page 2, line 12: maybe add some references here?

We added references Huner, Öquist, & Sarhan (1998).

 Page 5, line 25: not very clear why here a ÌE*psp is weighted to FRRF excitation LED, not in situ light?

This was done because sigma_PSII, measured by FRRF, is specific to the FRRF excitation LED wavelengths (470 nm) used in this instrument. When estimating 1/nPSII from sigma_PSII and a*psp, both of these measurements have to be specific to light of the same wavelengths. One can therefore either weight both to in situ light, or weight a*psp to FRRF excitation LED. The result would be the same.

Page 5, line 26: using assumption that ratio of PSII: PSI =1 whether will affect the accuracy of nPSII calcuation? especially for those samples under iron limitation, which should have decreased PSI abundance. Can authors provide the general range of PSII:PSI for samples with/without iron stress?

The manuscript actually does not state that we assume PSII:PSI to be equal.  Rather, we assume that 50% of absorbed energy goes to RCII, which does not imply a 1:1 ratio of the two reaction centers.

The referee is correct in pointing out that iron limitation can affect PSII:PSI ratios significantly. For example (Strzepek and Harrison, 2004) show that open ocean diatom PSII:PSI ratios can be as low as 11:1, while coastal diatoms show ratios of 2:1. This does not imply that in the open ocean diatom 10 times more of the absorbed energy will end up at RCII, when compared to RCI. This would be a serious problem, given that the two reaction centers work in series.

The differential allocation of absorbed light energy between RCII and RCI at different wavelengths can be estimated by comparing a*psp spectra and 730 nm fluorescence excitation spectra. The approach, explained in more detail in Suggett et al. (2004), relies on the fact that fluorescence is generally only emitted from pigments associated with PSII. Using this approach, Suggett et al. show that differential allocation of absorbed energy depends on taxonomy rather than growths conditions, and does not vary by more than 20%.

We added a reference to the review by (Kromkamp and Forster, 2003), which advises the use of 50% energy partitioning between PSII and PSI, as well as a sentence pointing out the potential error introduced by this assumption.

Page 6, line16: I feel eq.2 is very hard to follow, here are some questions

1) where is the Eis(λ) in Eq.(2) ?

   E_{IS}(λ) was present in Eq.(2), which has been copied below from the original manuscript.  We are not sure what the reviewer means.

$$E_{IS} = E_{LED} \cdot \frac{\sum_{400}^{700} a_{phy}(\lambda)\, E_{LED}(\lambda) \cdot \sum_{400}^{700} E_{IS}(\lambda)}{\sum_{400}^{700} a_{phy}(\lambda)\, E_{IS}(\lambda) \cdot \sum_{400}^{700} E_{LED}(\lambda)}$$

2) Maybe I missed somewhere but I cannot find where you mention that Eis(λ) (i.e. Eis at each wavelength) was measured? or you measured E0+(λ) ?, then using it to estimate Eis(λ). Sorry, just cannot find the related information.

The spectral distribution of light at 5 m depths ($E_{IS}(\lambda)$) was estimated as described in Schuback et al. (2016, 2017). This information was provided in the original manuscript two lines above Eq. (2).

3) not very clear why absolute values of light intensity for 14C P-E curve need to be corrected? And how can you correct light? I think you can only correct 14C-uptake rates, because C uptake rate measured under indoor LED light may differ with that under in situ natural light

The point is that we corrected the absolute value of PAR (i.e. light intensity integrated over the spectral range 400-700 nm) before fitting PI curves (i.e. modifying the x axis). This results in different curve fit parameters for light-dependent 14C uptake.

Page 6, line24: a little confused that why ˇ Φc-max = α*-14C / Ä ¸A*phy? is not =Pmax-14C

/ Ä ¸A*phy ?

We understand that this result may seem a little counter-intuitive. However, as discussed throughout the manuscript, the maximum efficiency of photosynthesis is achieved under light limitation. We have further emphasised this point in the revised version of the manuscript.

Page 6, line 20-25: I would suggest authors adding equations for how to calculate of ETR per second, but C-uptake is per hour. And it is same to ΦC. I think it will help to understand the meaning of Φe,C ΦC if authors can provide equations and parameters with unit here

We agree with the referee and have added equations 3, 4, 5, and 6.

Page 7, line8-9: I think the datasets Graff (2015) used for developing their bbp-Cphyto algorism mostly came from Open Ocean, where the phytoplankton is the main particle; however, when it is not the case (usually refer to Case II water), I think the algorism may not be suitable here, unless in this study area the backscatter signal mainly come from phytoplankton. And also, the author didn't provide the description of how they correct the backscatter data, so I would suggest authors to remove the phytoplankton carbon part.

The sentence has been removed.

Page 7, line 25, table 2: previously I thought NPQ should be highly correlated with surface PAR, but actually from the results in table 2 we can found oblivious "decoupling" exists within these two parameters. For example the second 24 hours 20:00, when the PAR is only 24, the NPQ value is actually higher than the NPQ at first 24 hours 12:00, when the PAR is 1054, do authors know the reason?

We apologize, the high NPQ(500) value given for the time-point 20:00 on the second day in table 2 was a typo. Nonetheless, the correct value, 2.23, is still higher than the value recorded for the 12:00 time-point on the first day (1.89), as the referee points out correctly.

Hysteresis is a common phenomenon in all aspects of photosynthesis at the molecular level, in particular on the diurnal scale. The fact that the potential for NPQ is higher at 20:00 on day 2 than at 12:00 on day one can be very simply explained by the daily photon-dose received by the phytoplankton at this point (See Figure 3a for reference).

Page 11, line 8: can you explain the reason of what may potentially cause mid-day ETR at OSP14 exceeds the maximum theoretical value

The important point here is that mid-day ETR exceeds the maximum level of **linear** electron transport. The fate of electrons not used for linear electron transport is described in some detail in the paragraph just following the sentence in question.

Page 11, line 8: as authors mentioned, the weak part of this MS is figure 7c, which is not very easy for primary productivity people to understand. It is telling that at OSP14, even water dominated by smaller phytoplankton and has nutrient limitation; it still has higher PB, which I think against most of the primary production research. Although it might be explained by NPP/GPP reason, I suggest in the future study authors should try to give or adjust the primary productivity rate to same level, for example, also measure respiration rate at same time.

We thank the referee for this suggestion.

Page 11, line 26:-27 adding some references here?

This statement is not so much based on references, but on the data presented in the present and previous studies. We added "We argue based on the results present here, and in Schuback et al. (2014; 2015; 2017),…" in front of the sentence, to clarify this further.

technical corrections

Page 2 line 25: "we examined diurnal variability. . .. . ."

Corrected.

Page 3, line 29: I cannot find Burt et al. (2018) in references.

(Burt et al., 2018)

Corrected.

Page 7, line 3 and Page 11, line 4: typing errors, "NQP" should be "NPQ"

Corrected.

Page 11, line 11 and 14, should be Fig 7d

Corrected.

Page 11, line 15, Fig 7e is missing here

Corrected.

Page 23, Figure 2: typing errors, "OSM14"in figure should be "OSP14"

Corrected.

Page 24, Table 1: method column, fourth item, ". . .. . .weighted to spectral distribution of in situ light"

Corrected.

Page 32 Figure 7: missing x-axis label

Corrected.

Page 28, 32, Figure 4 and 7. The unit for 14C-uptake should be per hour, not per second

Corrected.

References

Burt, W. J., Westberry, T. K., Behrenfeld, M. J., Zeng, C., Izett, R. W. and Tortell, P. D.: Carbon : Chlorophyll ratios and net primary productivity of Subarctic Pacific surface waters derived from autonomous shipboard sensors, Global Biogeochem. Cycles, doi:10.1002/2017GB005783, 2018.

Huner, N. P. ., Öquist, G. and Sarhan, F.: Energy balance and acclimation to light and cold, Trends Plant Sci., 3(6), 224–230, doi:10.1016/S1360-1385(98)01248-5, 1998.

Kromkamp, J. C. and Forster, R. M.: The use of variable fluorescence measurements in aquatic ecosystems: Differences between multiple and single turnover measuring protocols and suggested terminology, Eur. J. Phycol., 38(2), 103–112, doi:10.1080/0967026031000094094, 2003.

Strzepek, R. F. and Harrison, P. J.: Photosynthetic architecture differs in coastal and oceanic diatoms, Nature, 431(7009), 689–692, doi:10.1038/nature02954, 2004.

Suggett, D. J., MacIntyre, H. L. and Geider, R. J.: Evaluation of biophysical and optical determinations of light absorption by photosystem II in phytoplankton, Limnol. Oceanogr. Methods, 2(10), 316–332, doi:10.4319/lom.2004.2.316, 2004.

---

## Referee Comment (RC2) · Anonymous Referee #2 · 19 Feb 2019

This is an interesting and well-written paper by Schuback & Tortell that examines diurnal variability in photo-physiological characteristics of phytoplankton across two distinct environments. The study follows on from previous work under iron-limited conditions (Schuback et al. 2016) which is both interesting and well-cited in itself (21 citations at the time of review). In doing so, the authors provide additional insight into the effects of Fe-limitation upon primary productivity (PP), making this a nice comparison, and importantly, advance our capacity to apply Fast Repetition Rate fluorometry (FRRf) to measure PP through improved knowledge of $\Phi e,C$. Critically, the study addresses the

recently-established link between NSV and $\Phi e,C$ in further detail - as this is probably the most significant development in FRRf research for some time, further empirical evaluation of this relationship is exactly the direction this field should be heading in right now. Certainly, the study is both useful and timely (SCOR working group 156 highlights the current significance of this work) and thus merits publication. The methodology and data analysis appear robust, and the figures/tables are of a standard suitable for publication in this journal. Asides from once concern over the discussion of C-lifetimes (which can be easily addressed), I have no major issues with this manuscript and note that Reviewer 1 has done a thorough job pointing out many of the minor issues which I will avoid duplicating here.

I have however listed remaining minor issues that could be addressed to improve the manuscript further:

Carbon lifetime issue: Pg 6 Ln 5-7 (see also Pgs 8, 11) The authors make an important point about C-lifetimes, however I think they may have this slightly mixed-up (?). As I understand it, fast growing cells allocate more C to temporary storage which actually has a long-half time – therefore in a population of fast-growing cells, short 14C incubations measure something closer to gross PP, and for slow-growing cells, short incubations measure something closer to net PP. In this study (nutrient-limited), I think a 2 hr incubation would likely measure closer to net rather than gross (as stated). If so, this just needs correcting in text as it does not affect their interpretation of the data (as the authors correctly point out). I note that discussion of these trends occurs at three points in the manuscript and therefore minor text amendments may needed in each of these sections

Minor issues:

Pg 2 Ln 12-23 These reactions, operating on vastly different time scales, are ultimately powered by solar energy and are critically dependent on nutrient availability. (or critically depend)?

Pg 3 Ln 7-9 Some hyphens could be used here: "high-productivity, recently-published"

General Comment: Perhaps nit-picking here but I prefer to see the prime notation (′) used to denote light-acclimated parameters rather than what appears to be an apostrophe (') as used in this manuscript.

Pg 5 Ln 18. The authors make a great point about achieving a fully dark-regulated state required for ChlF parameters. Is this something that needs to be considered more in derivation of NSV? I really like the discussion surrounding possible explanations for the differences in slopes between NSV and Φe,C, and I wonder if an extra sentence or two could be included here to discuss whether the dark-regulated state is also a possible explanatory factor (Low initial Fv/Fm presumably = higher NSV?).

Pg 9 Ln 30 Typo: "prymnesiohytes" should be "prymnesiophytes"

Ln 22 "photoprotective" should be "photo-protective" (for consistency with surrounding text) Ln 28 "photophysiology" should be "photo-physiology"
* * *

---

## Author Comment (AC2) · 24 Feb 2019

This is an interesting and well-written paper by Schuback & Tortell that examines diurnal variability in photo-physiological characteristics of phytoplankton across two distinct environments. The study follows on from previous work under iron-limited conditions (Schuback et al. 2016) which is both interesting and well-cited in itself (21 citations at the time of review). In doing so, the authors provide additional insight into the effects of Fe-limitation upon primary productivity (PP), making this a nice comparison, and importantly, advance our capacity to apply Fast Repetition Rate fluorometry (FRRf) to measure PP through improved knowledge of $\Phi e,C$. Critically, the study addresses the recently-established link between NSV and $\Phi e,C$ in further detail - as this is probably the most significant development in FRRf research for some time, further empirical evaluation of this relationship is exactly the direction this field should be heading in right now. Certainly, the study is both useful and timely (SCOR working group 156 highlights the current significance of this work) and thus merits publication. The methodology and data analysis appear robust, and the figures/tables are of a standard suitable for publication in this journal. Asides from one concern over the discussion of C-lifetimes (which can be easily addressed), I have no major issues with this manuscript and note that Reviewer 1 has done a thorough job pointing out many of the minor issues which I will avoid duplicating here. I have however listed remaining minor issues that could be addressed to improve the manuscript further.

We thank referee #2 for their kind comments, and address all issues raised directly in the comments below. Page and line reference are given specific to the final, revised manuscript.

Carbon lifetime issue: Pg 6 Ln 5-7 (see also Pgs 8, 11) The authors make an important point about C-lifetimes, however I think they may have this slightly mixed-up (?). As I understand it, fast growing cells allocate more C to temporary storage which actually has a long-half time – therefore in a population of fast-growing cells, short 14C incubations measure something closer to gross PP, and for slow-growing cells, short incubations measure something closer to net PP. In this study (nutrient-limited), I think a 2 hr incubation would likely measure closer to net rather than gross (as stated). If so, this just needs correcting in text as it does not affect their interpretation of the data (as the authors correctly point out). I note that discussion of these trends occurs at three points in the manuscript and therefore minor text amendments may needed in each of these sections

We thank the referee for pointing out this mix-up and have corrected the text accordingly (P6 LN8-16; P8 LN35; P11 LN25-27).

Minor issues:

Pg 2 Ln 12-23 These reactions, operating on vastly different time scales, are ultimately powered by solar energy and are critically dependent on nutrient availability. (or critically depend)?

Corrected (P2 LN12).

Pg 3 Ln 7-9 Some hyphens could be used here: "high-productivity, recently-published" General Comment: Perhaps nit-picking here but I prefer to see the prime notation (â$\check{A}$š) used to denote light-acclimated parameters rather than what appears to be an apostrophe (') as used in this manuscript.

We agree and changed the annotation throughout the manuscript.

Pg 5 Ln 18. The authors make a great point about achieving a fully dark-regulated state required for ChlF parameters. Is this something that needs to be considered more in derivation of NSV? I really

like the discussion surrounding possible explanations for the differences in slopes between NSV and Φe,C, and I wonder if an extra sentence or two could be included here to discuss whether the dark-regulated state is also a possible explanatory factor (Low initial Fv/Fm presumably = higher NSV?).

We agree with the referee that under the iron-limited conditions at OSP14 it is less likely that a fully dark-regulated state (i.e. complete relaxation of all NPQ processes) was achieved for the noon samples. However, we do not think that this would affect the NPQ$_{NSV}$ values derived for in situ light availabilities, or the light-dependent increase in NPQ, which do not necessarily depend on the achievement of a dark-regulated state. Note that in Figure 8, only values derived for daylight hours are shown and used to derive the correlation.

Pg 9 Ln 30 Typo: "prymnesiohytes" should be "prymnesiophytes".

Corrected.

Ln 22 "photoprotective" should be "photo-protective" (for consistency with surrounding text).

Corrected.

Ln 28 "photophysiology" should be "photo-physiology".

Corrected.

---

## Author Response (AR2)

Dear Koji Suzuki,

We thank the editor and reviewers for their constructive comments and suggestions. All changes have been incorporated into the final version of the manuscript.

Best wishes,

Nina Schuback